*Report*

EMBO
Molecular Medicine

# Non-invasive lung cancer diagnosis by detection of *GATA6* and *NKX2-1* isoforms in exhaled breath condensate

Aditi Mehta[1,†], Julio Cordero[1,†], Stephanie Dobersch[1], Addi J Romero-Olmedo[1,2], Rajkumar Savai[3,4,§], Johannes Bodner[5], Cho-Ming Chao[6], Ludger Fink[7,§], Ernesto Guzmán-Díaz[8], Indrabahadur Singh[1], Gergana Dobreva[9], Ulf R Rapp[3,§], Stefan Günther[10], Olga N Ilinskaya[11], Saverio Bellusci[6,11,§], Reinhard H Dammann[12,§], Thomas Braun[10,§], Werner Seeger[3,4,§], Stefan Gattenlöhner[13], Achim Tresch[14,15], Andreas Günther[4,16,§] & Guillermo Barreto[1,11,*,§]

## Abstract

Lung cancer (LC) is the leading cause of cancer-related deaths worldwide. Early LC diagnosis is crucial to reduce the high case fatality rate of this disease. In this case–control study, we developed an accurate LC diagnosis test using retrospectively collected formalin-fixed paraffin-embedded (FFPE) human lung tissues and prospectively collected exhaled breath condensates (EBCs). Following international guidelines for diagnostic methods with clinical application, reproducible standard operating procedures (SOP) were established for every step comprising our LC diagnosis method. We analyzed the expression of distinct mRNAs expressed from *GATA6* and *NKX2-1*, key regulators of lung development. The Em/Ad expression ratios of *GATA6* and *NKX2-1* detected in EBCs were combined using linear kernel support vector machines (SVM) into the LC score, which can be used for LC detection. LC score-based diagnosis achieved a high performance in an independent validation cohort. We propose our method as a non-invasive, accurate, and low-price option to complement the success of computed tomography imaging (CT) and chest X-ray (CXR) for LC diagnosis.

**Keywords** EBC; *GATA6*; lung cancer; molecular diagnostics; *NKX2-1*
**Subject Categories** Biomarkers & Diagnostic Imaging; Cancer; Respiratory System

## Introduction

Lung cancer patients are mostly asymptomatic at early stages. Symptoms, even when present, are non-specific and mimic more common benign etiologies (Hyde & Hyde, 1974). Thus, in the majority of LC patients, traditional diagnostic strategies are initiated at advanced stages of the disease, when the overall condition of the patient is already impaired and prognosis is poor, as shown by the low 5-year patient survival of 1–5% (Herbst *et al*, 2008). Computed tomography imaging (CT) detects LC at an earlier stage than chest X-ray (CXR) (Henschke *et al*, 1999). Approximately 55–85% of the CT-detected LC can be surgically removed resulting in an improved 5-year patient survival of almost 52% (The International Early Lung

1 LOEWE Research Group Lung Cancer Epigenetic, Max Planck Institute for Heart and Lung Research, Bad Nauheim, Germany
2 Facultad de Ciencias Químicas, Universidad Autonoma "Benito Juarez" de Oaxaca, Oaxaca, Mexico
3 Department of Lung Development and Remodeling, Max Planck Institute for Heart and Lung Research, Bad Nauheim, Germany
4 Pulmonary and Critical Care Medicine, Department of Internal Medicine, Justus Liebig University, Giessen, Germany
5 Section Thoracic Surgery, Justus Liebig University, Giessen, Germany
6 Chair for Lung Matrix Remodeling, Excellence Cluster Cardio Pulmonary System, Justus Liebig University, Giessen, Germany
7 Institute of Pathology and Cytology, UEGP, Wetzlar, Germany
8 Regional Hospital of High Specialties of Oaxaca (HRAEO), Oaxaca, Mexico
9 Emmy Noether Research Group Origin of Cardiac Cell Lineages, Max Planck Institute for Heart and Lung Research, Bad Nauheim, Germany
10 Department of Cardiac Development and Remodeling, Max Planck Institute for Heart and Lung Research, Bad Nauheim, Germany
11 Institute of Fundamental Medicine and Biology, Kazan (Volga Region) Federal University, Kazan, Russian Federation
12 Institute for Genetics, Justus Liebig University, Giessen, Germany
13 Institute for Pathology, Justus Liebig University, Giessen, Germany
14 Max Planck Institute for Plant Breeding Research, Cologne, Germany
15 University of Cologne, Cologne, Germany
16 Agaplesion Lung Clinic Waldhof Elgershausen, Greifenstein, Germany
  *Corresponding author. Tel: +49 6032 705259; E-mail: guillermo.barreto@mpi-bn.mpg.de
  §Member of the Universities of Giessen and Marburg Lung Center (UGMLC) and the German Center of Lung Research (Deutsches Zentrum für Lungenforschung, DZL)
  †These authors contributed equally to this work

Cancer Action Program Investigators, 2006). Furthermore, the National Lung Cancer Screening Trial (NLST) reported a 20% decrease in LC mortality in the low-dose CT group (The National Lung Screening Trial Research Team, 2011). These studies support that early diagnosis is crucial to reduce the extremely high case fatality rate of LC (95%) (Ferlay *et al*, 2015). A better understanding of the molecular mechanisms involved in tumor initiation is critical to develop new diagnostic strategies for early LC diagnosis. We speculated that mechanisms involved in embryonic development are recapitulated during LC initiation (Borczuk *et al*, 2003; Liu *et al*, 2006). Thus, GATA6 (GATA-binding factor 6) and NKX2-1 (NK2 homeobox 1, also known as TTF-1, thyroid transcription factor-1), two transcription factors that are key regulators of embryonic lung development (Maeda *et al*, 2007), were analyzed for their potential as biomarkers for LC detection. Among other lung development relevant genes, *GATA6* and *NKX2-1* were selected due to their similar gene structure (Fig 1A) and their implication in LC (Winslow *et al*, 2011; Cheung *et al*, 2013). We hypothesized that isoform-specific expression analysis of *GATA6* and *NKX2-1* in EBCs can be used for LC diagnosis. Further, we established reproducible SOPs for a LC diagnosis method following international guidelines for diagnostic methods with clinical application (Bossuyt *et al*, 2003; McShane *et al*, 2013).

## Results

### Study design

A detailed description of the study population can be found in the Materials and Methods section, Tables 1 and 2. The study population consisted of two types of samples, retrospectively collected FFPE human lung tissue samples and prospectively collected EBCs. The study population was grouped in three sets according to the phase of the study in which the samples were analyzed (Fig EV1). In the first phase, the SOPs for RNA isolation and qRT–PCR analysis were established using retrospectively collected FFPE human lung tissue samples ($n = 112$, Table 1) collected in Germany and Mexico. Further, isoform-specific expression analysis of *GATA6* and *NKX2-1* was performed on these FFPE samples using the optimized SOP. In the second phase, the SOP for collection, storage, and processing of EBCs was optimized. Using this SOP, a training set of EBCs ($n = 113$, Table 1) was prospectively collected. Isoform-specific expression analysis of *GATA6* and *NKX2-1* was performed on the training set. Next, a SVM classifier was used to combine the Em/Ad ratios of *GATA6* and *NKX2-1* of each sample to create the LC score. In the third phase, using the SOPs established in phases 1 and 2, an independent set of previously unseen EBCs (validation set, $n = 138$, Table 1) was collected from patients continuously enrolled in the clinic, in a blinded manner, thereby mimicking conditions of clinical use. Isoform-specific expression analysis of *GATA6* and *NKX2-1* was performed on the validation set. The LC score was externally validated on this set of EBCs.

### Expression ratios of Em to Ad isoforms of *GATA6* and *NKX2-1* as biomarkers for LC

*In silico* analysis of *GATA6* and *NKX2-1* revealed a common gene structure (Fig 1A, top). Two promoters were predicted in each of

the genes, one 5′ of the first exon and the other one in the first intron. Expression analysis showed that each gene gave rise to two distinct transcripts (bottom) driven by different promoters. The structure of the murine orthologues is similar to humans, highlighting an evolutionarily conserved gene structure. Expression analysis by qRT–PCR during mouse lung development showed that the expression of both isoforms of the same gene was complementary and differentially regulated (Fig EV2A), with the embryonic (Em) isoform mainly expressed during early developmental stages, and the adult (Ad) isoform expressed at later stages and in the adult lung. Interestingly, the expression of the Em isoforms of *GATA6* and *NKX2-1* was higher than the expression of the Ad isoforms in different LC cell lines when compared to human control lung tissue (Fig EV2B). We confirmed these results in various mouse models of LC (Fig EV2C and Appendix Supplementary Results).

To verify that a similar increase in the expression of the Em isoforms of *GATA6* and *NKX2-1* occurs in LC patients, we performed isoform-specific expression analysis in human lung tissues from controls and LC patients (Fig 1B). In order to minimize the effect of individual variations among the different LC specimens and to facilitate comparability between LC and control samples, we used the expression of the Ad isoform as internal control for a second level of normalization in addition to the normalization using the endogenous reference gene *TUBA1A* (Mehta *et al*, 2015a). Thus, we calculated the Em/Ad expression ratio (Em/Ad) for each sample. In control lung tissue ($n = 61$), the median Em/Ad was 0.15 for *GATA6* and 0.18 for *NKX2-1* (Table EV1). Interestingly, the median Em/Ad increased in the LC tissue ($n = 51$) to 2.25 ($P = 4.1E\text{-}16$) for *GATA6* and to 1.62 ($P = 3.1E\text{-}18$) for *NKX2-1,* consistent with our results in Fig EV2. The increased Em/Ad expression ratios of *GATA6* and *NKX2-1* in the LC samples were maintained after sample grouping by ethnicity or gender (Fig 1C and Table EV1). Further, sample grouping based on the TNM classification (Sobin *et al*, 2009) revealed that the Em/Ad expression ratios increased to 2.23 ($P = 4.6E\text{-}10$) for *GATA6* and 1.67 ($P = 6.3E\text{-}3$) for *NKX2-1* at stage I (Fig 1D and Table EV2), supporting the use of increased Em/Ad of *GATA6* and *NKX2-1* as biomarkers for detection of early staged LC.

### Detection of Em and Ad isoforms of *GATA6* and *NKX2-1* in EBC

After confirming that RNA-containing exosomes are enriched in EBCs of LC patients (Fig EV2D and E, and Appendix Supplementary Results), we tested whether increased Em/Ad expression ratios of *GATA6* and *NKX2-1* can also be detected in EBCs. Thus, we performed isoform-specific expression analysis by qRT–PCR after total RNA isolation from EBCs (Fig 2A). The SOPs for EBC collection, storage, and processing were optimized (Fig EV3 and Appendix Supplementary Results) and subsequently used for the collection of a training set of EBCs ($n = 113$, Table 1). In control EBCs ($n = 65$), the median Em/Ad was 0.15 for *GATA6* and 0.23 for *NKX2-1* (Table EV1). Correlating with our previous results using lung tissues, the median Em/Ad increased in the EBCs of LC patients ($n = 48$) to 1.32 ($P = 2.26E\text{-}16$) for *GATA6* and to 1.13 ($P = 1.6E\text{-}17$) for *NKX2-1*. Hence, our results support that an increased Em/Ad of *GATA6* and *NKX2-1* can be also detected in EBCs from LC patients. The specificity of the different qRT–PCR products detected in the EBCs was demonstrated by different

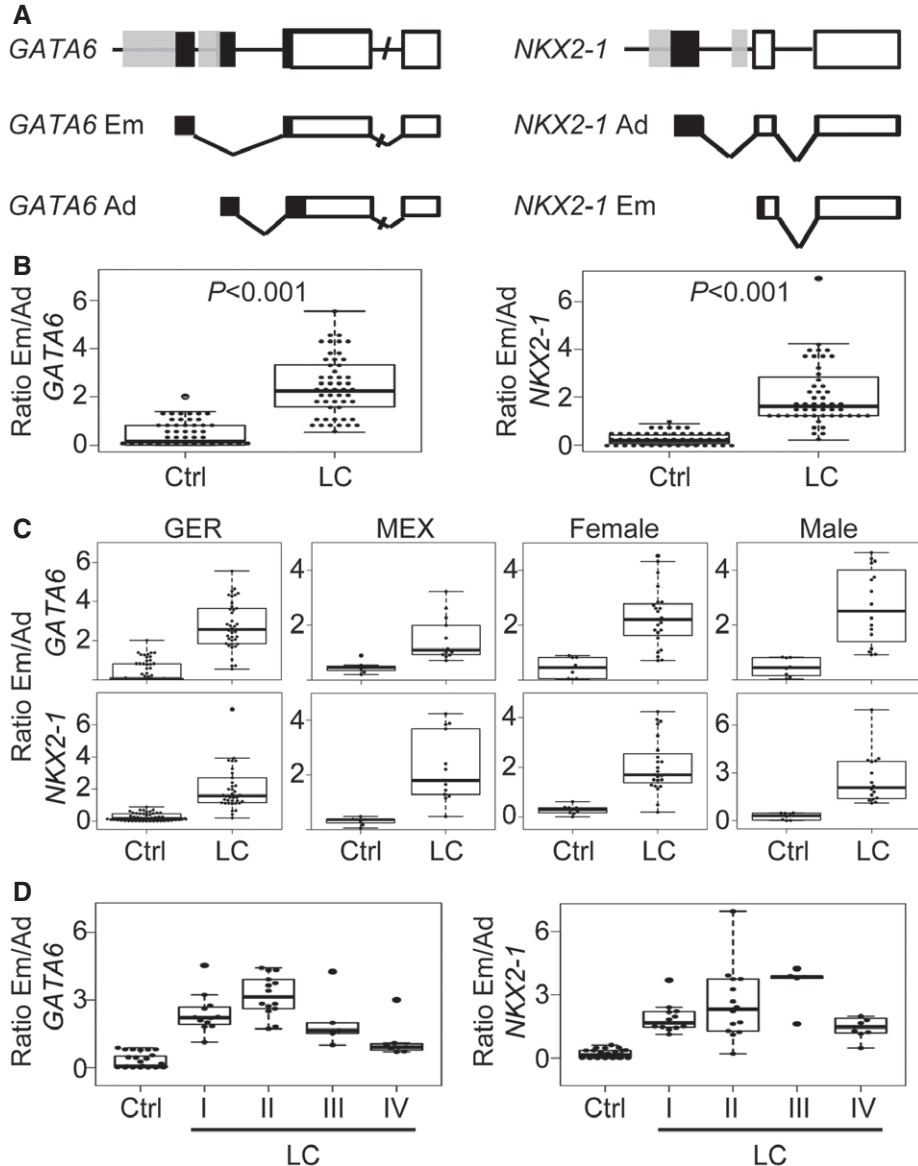

**Figure 1.  Em/Ad expression ratios of *GATA6* and *NKX2-1* as LC biomarkers.**

A   *In silico* analysis of human *GATA6* and *NKX2-1* shows similar gene structures (top) with two promoters (gray boxes) driving the expression of two distinct transcripts (middle and bottom); exons as black (non-coding) and white (coding) boxes. *GATA6*, GATA-binding factor 6; *NKX2-1*, NK2 homeobox 1; Em, embryonic; Ad, adult.

B   Box plots of the Em/Ad expression ratios of *GATA6* (left) and *NKX2-1* (right) in FFPE lung tissue sections from controls (Ctrl, *n* = 61) or lung cancer (LC, *n* = 51) patients. Isoform-specific expression of the indicated genes was analyzed by qRT–PCR after total RNA isolation from tissue samples.

C   Box plots of Em/Ad of *GATA6* (top) or *NKX2-1* (bottom) show that high Em/Ad ratios in LC samples are maintained among ethnic groups (left) and gender (right). GER, samples collected in Germany; MEX, samples collected in Mexico.

D   Box plots of Em/Ad of *GATA6* (left) or *NKX2-1* (right) in FFPE lung tissue sections from controls or LC patients. Samples were staged according to the TNM classification (Sobin *et al*, 2009).

Data information: Each point represents one sample. The five-number summary and the statistical test values from each plot are shown in Tables EV1 and EV2.

techniques (Appendix Figs S1 and S2). The repeatability of isoform-specific expression analysis of *GATA6* and *NKX2-1* was confirmed by Bland–Altman plots (Bland & Altman, 1986) after measurements in two distinct EBCs from the same patient, for five different patients (Fig EV4B and Appendix Supplementary Results). To further validate our findings, EBC-based expression analysis was directly compared with LC tissues of the same patient

(Fig EV4C). The Em/Ad expression ratios of *GATA6* and *NKX2-1* obtained from both types of samples of the same individuals showed a strong positive correlation ($R^2$ = 0.48 for *GATA6* and 0.69 for *NKX2-1*). Moreover, the classical methods for LC diagnosis, histopathology, and immunohistochemistry directly correlated with the increased Em/Ad of *GATA6* and *NKX2-1* in all cases that we tested.

**Table 1.   Classification of study population.**

| Classification | Pathological diagnosis | FFPE tissue samples | | | | Exhaled breath condensates | | | | | | | |
| | | | | | | Training set | | | | Validation set | | | |
| | | Number samples | Ethnicity | | Total | Number samples | Ethnicity | | Total | Number samples | Ethnicity | | Total |
| | | | GER | MEX | | | GER | MEX | | | GER | MEX | |
| Ctrl | Healthy donors | 27 | 19 | 8 | 61 | 22 | 19 | 3 | 65 | 30 | 29 | 1 | 78 |
| | COPD | 19 | 19 | 0 | | 10 | 10 | 0 | | 12 | 12 | 0 | |
| | IPF | 15 | 15 | 0 | | 33 | 33 | 0 | | 36 | 36 | 0 | |
| NSCLC | ADC | 17 | 11 | 6 | 50 | 24 | 18 | 6 | 42 | 13 | 13 | 0 | 57 |
| | SQCC | 12 | 9 | 3 | | 10 | 9 | 1 | | 9 | 9 | 0 | |
| | LCC | 2 | 1 | 1 | | 1 | 0 | 1 | | 0 | 0 | 0 | |
| | ASCC | 1 | 0 | 1 | | 1 | 0 | 1 | | 0 | 0 | 0 | |
| | NS | 18 | 16 | 2 | | 6 | 6 | 0 | | 35 | 35 | 0 | |
| SCLC | | 1 | 0 | 1 | 1 | 6 | 5 | 1 | 6 | 3 | 3 | 0 | 3 |
| Total number of samples | | 112 | 90 | 22 | 112 | 113 | 100 | 13 | 113 | 138 | 137 | 1 | 138 |

FFPE, formalin-fixed and paraffin-embedded tissue samples; GER, samples collected in Germany; MEX, samples collected in Mexico; Ctrl, control; NSCLC, non-small cell lung cancer; ADC, adenocarcinoma; SQCC, squamous cell carcinoma; LCC, large cell carcinoma; ASCC, adeno-squamous cell carcinoma; SCLC, small cell lung cancer; COPD, chronic obstructive pulmonary disease; IPF, idiopathic pulmonary fibrosis; NS, non-specified. Pathological diagnosis is according to the current diagnostic criteria for morphology, immunohistochemistry, and genetic findings.

**Development and validation of the LC score as a simple clinical score for LC diagnosis**

Our results demonstrated that the Em/Ad expression ratios of *GATA6* and *NKX2-1* in the EBCs of LC patients can be used for LC detection. Nonetheless, we decided to improve our method by combining the log2-transformed Em/Ad expression ratios for *GATA6* and *NKX2-1* of each sample of the training set of EBCs using a SVM (Fig 2B). The SVM calculated a robust separating hyperplane (left), and the distance of each point to this hyperplane is the LC score. A sample with a LC score greater than zero is classified as a LC patient (right; Appendix Table S2), while samples with LC score equal or less than zero are classified as control. To further confirm the performance of the LC score classifier, we conducted an external validation (Fig 2C and Appendix Table S3) on an independent set of EBCs (validation set; $n$ = 138) that was collected from patients continuously enrolled in the clinic, in a blinded manner. Receiver operating characteristic curve (ROC) analysis (Sing *et al*, 2005) confirmed the increase in performance of the LC score-based classification over a classification relying only on Em/Ad expression ratios of *GATA6* or *NKX2-1* (Fig 2D). The area under the curve (AUC) values were 0.99 (LC score), 0.93 (*GATA6*), and 0.97 (*NKX2-1*). Further performance assessment of the LC score after applying it to the independent validation set of EBCs showed a sensitivity of 98.3%, and a specificity of 89.7% (Fig 2E and Table EV4).

Cigarette smoking is strongly associated with LC (Mehta *et al*, 2015b). Thus, we decided to assess whether the LC score reflected the smoking history of the individuals (Fig 3A). Controls and LC samples were sorted into three groups: never smokers (NS), previous smokers (PS), and current smokers (CS). The LC score was significantly different between Ctrl and LC patients in each of the smoking history groups (Table EV3). However, we did not find significant differences between the smoking history groups neither in the Ctrl nor in the LC group. Further, using the LC score, we were unable to statistically discriminate between the two major LC types NSCLC and SCLC (Fig 3B and Table EV2) or different histological subtypes of NSCLC (ADC, SQCC or LCC). Remarkably, sample grouping based on TNM classification (Sobin *et al*, 2009) (Fig 3C and Table EV2) revealed that the median LC score increased from −3.66 in the controls to 1.52 ($P$ = 1E-6) and 2.89 ($P$ = 1E-6) in EBCs from patients with LC at stages I and II, respectively, demonstrating the potential of our method for the detection of stage I/II LC.

## Discussion

Breath capture methods for diagnostic purposes range from directly breathing into a highly sensitive analysis platform (electronic Nose, eNose) (Mehta *et al*, 2015b) or the relatively simple collection of exhaled breath through cooling devices as proposed here. Recent studies have reported the use of EBC for the detection of DNA mutations and DNA methylation patterns in LC patients (Dent *et al*, 2013; Xiao *et al*, 2014). However, there are some discrepancies between different reports that can be explained as EBCs are highly diluted mixtures of compounds. Thus, EBC-based LC diagnosis requires appropriate stringent standardization protocols in order to reduce variability and increase sensitivity of the technique (Horvath *et al*, 2005). Consistent with this line of reasoning, we established strict SOPs for EBC collection, storage, and processing for isoform-specific expression analysis. Our work demonstrated that RNA purified from EBC can be used for qRT–PCR-based isoform-specific expression analysis of *GATA6* and *NKX2-1*, despite the relatively high fragmentation of the isolated RNA (Appendix Supplementary Results). Similarly, qRT–PCR-based expression of genes has been successfully

**Table 2.   Clinical characteristics of patients with lung cancer.**

| Clinical characteristic | % Samples | | | | |
| | FFPE samples | EBC | | | |
| | | Training set | | Validation Set | |
| | | Ctrl | LC | Ctrl | LC |
| Age | | | | | |
| < 50 | 8.3 | 9.1 | 8.3 | 4 | 3.7 |
| 50–70 | 58.4 | 60 | 50 | 60 | 66.6 |
| > 70 | 33.3 | 30.9 | 41.7 | 36 | 29.6 |
| Gender | | | | | |
| Male | 42.8 | 54 | 56 | 51 | 60 |
| Female | 57.1 | 46 | 44 | 49 | 40 |
| Ethnic group | | | | | |
| GER | 64.3 | 95.3 | 79.2 | 98.6 | 100 |
| MEX | 35.7 | 4.6 | 20.8 | 1.4 | 0 |
| Smoking history | | | | | |
| Current (CS) | 50 | 18.2 | 44.4 | 20 | 19.1 |
| Former (PS) | | 4.5 | 11.1 | 10 | 70.2 |
| Never (NS) | 50 | 77.2 | 44.4 | 70 | 10.7 |
| Stage[a] | | | | | |
| I | 30.77 | – | 14.7 | – | 14.8 |
| II | 35.9 | – | 17.6 | – | 18.5 |
| III | 12.8 | – | 38.3 | – | 59.2 |
| IV | 20.5 | – | 29.4 | – | 7.4 |
| Recurrent disease | 16.6 | – | 22.9 | – | 6.6 |
| Active treatment ongoing at sample collection[b] | 25.5 | – | 22.9 | – | 31.7 |

FFPE, formalin-fixed and paraffin-embedded tissue samples; EBC, exhaled breath condensate; samples collected in Germany (GER) and Mexico (MEX). Percentages may not exactly add up to 100% because of rounding.
[a]Samples were staged according to the TNM Classification (Sobin et al, 2009).
[b]Individuals undergoing active treatment, including chemo and/or radiation therapy, at the time of sample collection.

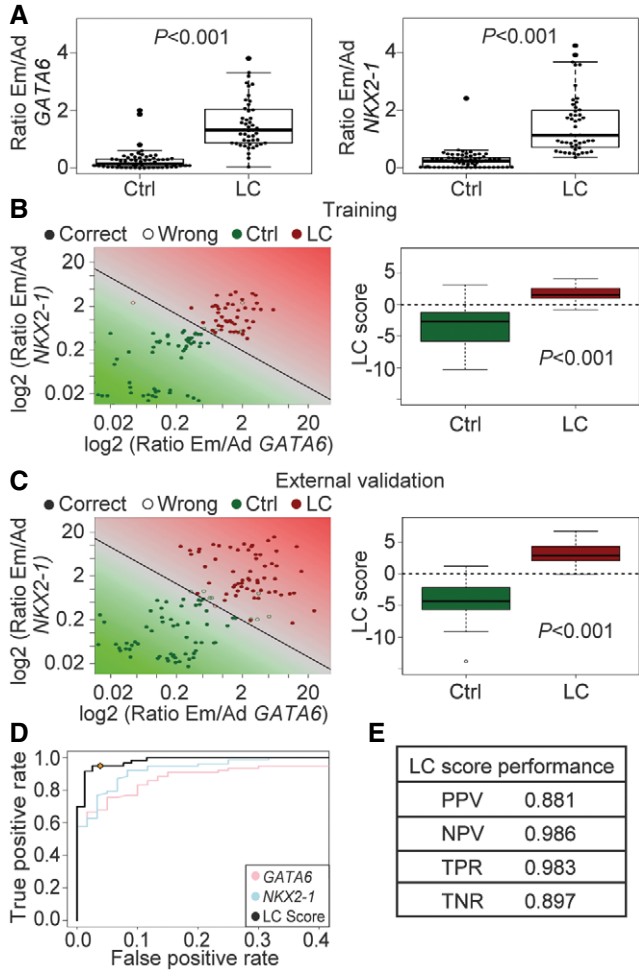

**Figure 2.   Development and validation of a simple clinical score for LC diagnosis.**

A   Box plots of the Em/Ad expression ratios of GATA6 (left) and NKX2-1 (right) in EBCs from controls (Ctrl, n = 65) or lung cancer (LC, n = 48) patients (training set). Each point represents one sample.

B   Learning of the SVM on the training set of EBCs. Left, the log2-transformed Em/Ad expression ratios of GATA6 (x-axis) and NKX2-1 (y-axis) detected in the EBCs of the training set were plotted. The SVM combined the Em/Ad of GATA6 and NKX2-1 of each sample and created a linear classifier. The solid line represents the decision boundary of this classifier, which reliably separates control (green dots) and LC (red dots) samples. Filled circles, correctly classified samples; open circles, wrongly classified samples. Right, box plot of the LC score on the training set of EBCs.

C   External validation of the LC score. Left, the log2-transformed Em/Ad expression ratios of GATA6 and NKX2-1 detected in an independent set of blinded collected EBCs were plotted as in (B). The SVM-based classifier confirmed its discriminatory power between control and LC samples. Right, box plot of the LC score on the validation set of EBCs.

D   ROC analysis confirmed the increase in performance of the LC score-based classification on the validation set of EBCs (black line) over a classification relying only on Em/Ad expression ratios of GATA6 (pink line) or NKX2-1 (blue line). The orange diamond represents the optimal operating point of the SVM classifier, that is, the point on the curve with maximal Youden's J index.

E   Performance of the LC score on the validation set of EBCs. PPV, prospective predictive value; NPV, negative predictive value; TPR, true positive rate or sensitivity; TNR, true negative rate or specificity.

Data information: The five-number summary and the statistical test values from each box plot are shown in Table EV1.

performed using highly fragmented RNA isolated from FFPE samples (Shane et al, 2010). Indeed, our SOPs for assay operation were initially optimized using RNA isolated from FFPE samples and subsequently applied to EBCs. The specificity of our assay conditions was demonstrated by sequencing the different qRT–PCR products detected in the EBCs (Appendix Figs S1 and S2). Interestingly, a sequence search in all public mRNA databases using AceView (http://www.ncbi.nlm.nih.gov/IEB/Research/Acembly/) revealed four different transcripts of NKX2-1 and two of GATA6, including the Em and Ad isoforms reported here. Further, the Em and Ad isoforms of NKX2-1 have been reported in lung tissue (Li et al, 2000) whereas the GATA6 isoforms were detected in other tissues (Brewer et al, 1999). Analysis of RNA-sequencing data deposited at The Cancer Genome Atlas (TCGA; http://cancergenome.nih.gov/) confirmed the increased expression of the Em isoform of GATA6 in LC samples when compared with control samples (Fig EV5). However,

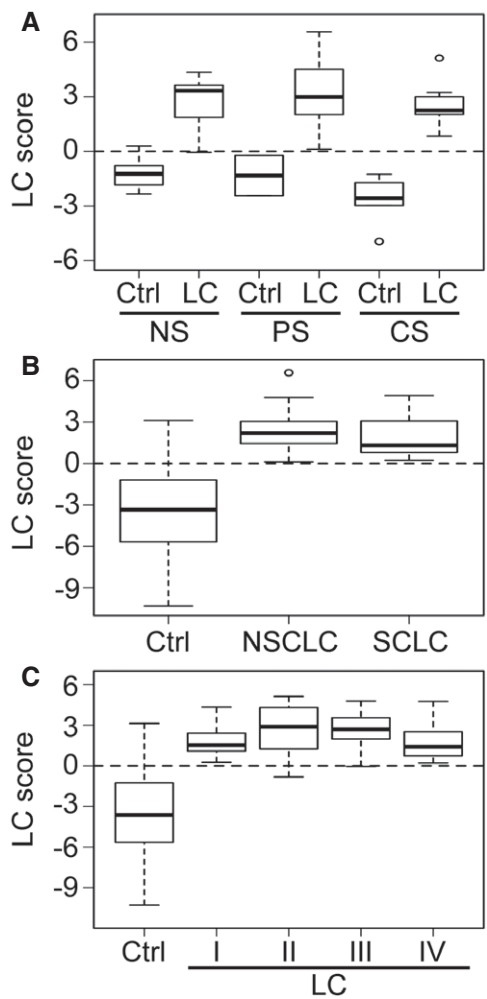

**Figure 3.  LC score can be used in EBCs for diagnosis of LC at early stages.**

A   LC score does not correlate with smoking history. Box plot of the LC score detected in EBCs from control (Ctrl) and lung cancer patients (LC). Samples were grouped based on the smoking history into never smokers (NS), previous smokers (PS), and current smokers (CS).

B   LC score does not discriminate between the major LC subtypes. Box plot of the LC scores detected in EBCs from Ctrl, non-small cell lung cancer (NSCLC), and small cell lung cancer (SCLC) patients.

C   LC score can be used for detection of early staged LC. Box plot of the LC scores detected in EBCs from Ctrl and LC patients of stages I, II, III, and IV. Patients were staged according to the TNM classification (Sobin *et al*, 2009).

Data information: The five-number summary and the statistical test values from each plot are shown in Tables EV1–EV3.

discrimination between both *NKX2-1* isoforms was not possible due to the data format available at the TCGA.

Our SVM-based LC score classifier was unable to discriminate between EBCs from patients with different NSCLC subtypes, thereby showing clear limitations of our approach that could be overcome in the future by extending the EBC-based expression analysis to known markers of different NSCLC subtypes. Immunodetection of NKX2-1 in combination with NAPSA and immunonegativity of TP63 is sufficient to distinguish ADC among the other NSCLC subtypes (Noh & Shim, 2012). In contrast, we detected increased expression of the Em isoform of *NKX2-1* in all three subtypes of NSCLC. A plausible

explanation for this discrepancy might be that transcript detection is more sensitive than immunodetection. Supporting this line of ideas, amplification of *NKX2-1* has been reported in 20% of 99 SQCC cases analyzed by FISH but not detected at the protein level (Tang *et al*, 2011) and 14q13.3 amplification, containing *NKX2-1*, is one of the most significant amplifications in SQCC reported in TCGA.

As described in the study population, IPF (idiopathic pulmonary fibrosis) and COPD (chronic obstructive pulmonary disease) samples were included into the control groups since they are non-malignant hyperproliferative lung diseases with an increased risk of LC (Turner *et al*, 2007; Li *et al*, 2014). Moreover, IPF and COPD are frequently found comorbidities in LC. Consistent with these findings, four of the eight wrongly classified control samples in the validation set were COPD samples (Appendix Table S3). Further, performance assessment values of the LC score using a control population with (Fig 2E) or without IPF and COPD samples (Table EV4) were similar, arguing against a potential bias by incorporating the IPF and COPD samples into the control population.

Even though our LC diagnosis method was externally validated on an independent set of EBCs, the results of our study are insufficient to safely predict its usefulness under clinical conditions, for which a suitably designed, large prospective study would be required. Despite the limitations of our study, the results presented here are promising, since our method is able to detect LC at stages I and II. We propose to incorporate our method into the current protocols for patients undergoing diagnostic evaluation for pulmonary diseases characterized by hyperproliferation. In addition, we suggest to integrate our technology into CT-based LC screening approaches in high risk populations (Colditz *et al*, 2000; Bach *et al*, 2003; Spitz *et al*, 2007, 2008; de Torres *et al*, 2007; Cassidy *et al*, 2008), a procedure routinely used in the USA, but not in Europe due to concerns regarding the very high percentage of false-positive observations (> 90%) and hence low specificity (73.4%) (The National Lung Screening Trial Research Team, 2013), resulting in unnecessary follow-up CT scans, bronchoscopy, or even surgery (Jett, 2005). Concomitant implementation of EBC-based LC detection together with CT could help to reduce the false-positive rate of CT imaging, for example, in cases with suspicious image findings, thereby preventing individuals from being unnecessarily exposed to high dose of radiation or surgery. Routine implementation of EBC-based molecular diagnosis may become an accurate, straightforward, non-invasive, and low-price option to complement the success of CT for LC diagnosis.

## Materials and Methods

### Study population

The study was performed according to the principles set out in the WMA Declaration of Helsinki and to the protocols approved by the institutional review board and ethical committee of Regional Hospital of High Specialties of Oaxaca (HRAEO), which belongs to the Ministry of Health in Mexico (HRAEO-CIC-CEI 006/13), and the Faculty of Medicine of the Justus Liebig University in Giessen, Germany (AZ.111/08-eurIPFreg). A flowchart depicting the three phases of the study and each step during the development of the

EBC-based LC diagnostic method is represented in Fig EV1. The study population (described in Tables 1 and 2) consisted of two types of samples, formalin-fixed paraffin-embedded (FFPE) human lung tissue, and exhaled breath condensates (EBCs). Samples were collected in two different cohorts located in Mexico and Germany, allowing us to investigate ethnic differences. All participants provided informed written consent.

During the first phase of the study, FFPE samples of either diagnostic transbronchial or bronchial biopsies or oncologic resections were retrospectively collected. All cases were reviewed and staged by an expert panel of pulmonologists and oncologists in the different cohorts according to the current diagnostic criteria for morphological features and immunophenotypes recommended by the International Union Against Cancer (Sobin et al, 2009). Inclusion criteria for the FFPE lung tissue samples were primary small cell lung cancer (SCLC) and non-small cell lung cancer (NSCLC) samples, including lung adenocarcinoma (ADC), squamous cell carcinoma (SQCC), large cell carcinoma (LCC), and adenosquamous carcinoma (ASCC; Table 1). Samples older than 5 years were excluded. FFPE tissue samples of LC patients comprised approximately 80% tumor cells. The control population for the analysis of FFPE samples included lung tissue that was taken from macroscopically healthy adjacent regions of the lung of LC patients and control lung tissue that was obtained from age-matched donor lungs, who had no diagnosis or family history of LC, in the frame of surgical size reduction of the donor lung during lung transplantation. Lung tissue samples from idiopathic pulmonary fibrosis (IPF) and chronic obstructive pulmonary disease (COPD) patients, both diagnosed according to international guidelines (www.goldcopd.org; Wuyts et al, 2012), were also included in to the control population because these lung diseases have been reported to increase LC risk when compared to individuals with normal pulmonary function (Turner et al, 2007; Li et al, 2014). In addition, the IPF and COPD cohorts were included in the study to determine the discriminatory power of the diagnosis method proposed here with respect to other non-cancer diseases characterized by alveolar or bronchiolar hyperproliferation.

During the second and third phases of the project, EBCs were prospectively collected in two chronologically separated sets, training set, and validation set. Different operators in different cohorts collected the EBCs using the optimized SOP thereby supporting feasibility for clinical implementation. Inclusion criteria for both sets of EBCs were that the patients were undergoing diagnostic evaluation for LC, IPF, or COPD (prior to transbronchial biopsy) at the Regional Hospital of High Specialties of Oaxaca (HRAEO), Mexico (from July to December 2013), the Agaplesion Lung Clinic Waldhof Elgershausen (from March 2014 to February 2015), and the Clinic of the Universities of Giessen and Marburg (from October 2014 to February 2015). The training set of EBCs was collected and used during the second phase of the project for the development of the LC score-based classifier and comprised 65 controls and 48 LC patients (Table 1). The control population consisted of EBCs from healthy donors (22 individuals with no symptoms, no complaints, and no prior history of LC or any other pulmonary disease), IPF, and COPD patients (33 and 10 EBCs, respectively). The rationale for including the IPF and COPD cohorts into control population was explained in the previous paragraph. The second set of EBCs (validation set, Table 1) was collected at a later time point with respect

to the training set and consisted of 78 independent controls and 60 previously unseen LC patients that were collected without prior knowledge of the clinical diagnosis (blinded sample collection) by different operators at different centers. The LC score-based classifier was applied to the validation set of EBCs for external validation and achieved high performance. The training and validation sets were comparable in their distribution of controls and LC samples, smoking history, age, and gender of the individuals (Table 2).

Within the LC groups of both types of samples, FFPE and EBCs, sample distribution correlated with the general LC epidemiology: the majority of the LC samples represented adenocarcinomas (ADC), followed by squamous cell carcinomas (SQCC; Table 1); the majority of the patients were in the age group of 50–70 years, were current or former smokers, and both male and female patients were equally represented (Table 2). Further, the majority of the patients were in the stages I–III of the disease and only a minority had a recurrent disease (Table 2). Importantly, the control groups were representative of the LC groups with respect to age and gender distribution (Table 2).

### Exhaled breath condensate collection

Exhaled breath condensate collection was performed using the RTube (Respiratory Research) as described online (http://www.respiratoryresearch.com/products-rtube-how.htm) and following the guidelines for EBC sampling by the ERS/ATS Task Force (Horvath et al, 2005). For EBC collection, it was advised that all donors refrain from eating and drinks (except water) for 2 h before EBC collection. Donors were awake and breathing normally without mechanical ventilation. Prior to EBC collection, donors were asked to rinse the mouth with freshwater to avoid any additional contaminants. Sample was collected with the Rtube using a nose clamp to avoid nasal contaminants, and breathing was only through the mouthpiece. For each donor, EBC collection was performed for 10 min of tidal breathing. However, if the donors felt any discomfort and/or inability to continue, a minimum time of 5 min was acceptable without any loss in quality of the material obtained. After EBC collection, the samples were stored immediately at −80°C in 500 μl aliquots. It is essential that the samples are frozen as soon as possible after EBC collection (Fig EV3E and F). The EBC was stored in microcentrifuge tubes that were treated with RNaseZap (Life technologies) and autoclaved twice. All steps during the collection and processing of EBCs were performed under RNase-free conditions, including the use of barrier–filter tips and cleaning all surfaces and gloves with RNaseZap, which are critical to ensure the integrity and quality of the samples.

### Cell culture and mouse experiments

In this study, we used human lung adenocarcinoma cell lines (A549; CCL-185 and A427; HTB-53) and a human bronchoalveolar carcinoma cell line (H322; CRL-5806). In addition, Mus musculus Lewis lung cancer cell line (LLC1; CRL-1642) was used in a mouse model of experimental metastasis, wherein 1 million LLC1 cells were injected into the tail vein of experimental mice. For the xenograft model, lung tumors were generated by intratracheal instillation of 2 million A549 cells into BALB/c nu/nu mice as described (Savai et al, 2009).

    

Cell lines were cultured in medium and conditions recommended by the American Type Culture Collection (ATCC). Cells were used for the preparation of RNA (QIAGEN RNeasy plus mini kit).

Five- to six-week-old C57BL6 and 7- to 8-week-old BALB/c nu/nu mice were used in this study. Animals were housed under controlled temperature and lighting [12/12-h light/dark cycle], fed with commercial animal feed and water *ad libitum*. For the mouse model of experimental metastasis, LLC1 cell suspension of 1 million cells/100 μl was prepared in sterile phosphate buffer saline (PBS). C57BL6 control mice ($n = 3$) were injected with 100 μl PBS whereas experimental mice ($n = 5$) with 100 μl of cell suspension into the tail vein of each mouse. The development of tumors was monitored 21 days post-injection. Lung tissue was harvested from each mouse separately for RNA isolation and isoform-specific expression analysis.

Mouse work was performed in compliance with the German Law for Welfare of Laboratory Animals. The permission to perform the experiments presented in this study was obtained from the Regional Council (Regierungspräsidium in Darmstadt, Germany). The numbers of the permissions are V54-19c20/15-B2/345; IVMr46-53r30.03.MPP04.12.02; and IVMr46-53r30.03.MPP06.12.01. Animals were killed for scientific purposes according to the law mentioned above which complies with national and international regulations.

### Gene expression analysis by qRT–PCR

Total RNA was isolated from cell lines using the RNeasy Mini kit (Qiagen). Human lung tissue samples were obtained as formalin-fixed paraffin-embedded (FFPE) tissues, and eight sections of 10-μm thickness were used for total RNA isolation using the RecoverAll™ Total Nucleic Acid Isolation Kit for FFPE (Ambion). Total RNA isolation from EBC was performed using 500 μl of sample and the RNeasy Micro Kit (Qiagen). Complementary DNA (cDNA) was synthetized using the High Capacity cDNA Reverse Transcription kit (Applied Biosystem) with 0.5–0.7 μg (EBC) or 1 μg (FFPE sample) total RNA. RT reaction without adding enzyme was used as negative control. qRT–PCRs were performed using SYBR® Green on the Step One plus Real-time PCR system (Applied Biosystems) using the primers specified in the Appendix Table S1. Briefly, 1× concentration of the SYBR Green master mix, 250 nM each forward and reverse primer, and 3.5 μl (EBC) or 1 μl (cell lines, mice and human lung cancer tissue) from a sixfold diluted RT reaction were used for the gene-specific qPCR.

The integrity of isolated mRNA and the performance of the RT reaction were determined by ratios of expression of the housekeeping genes *GAPDH* and *HPRT1* using two different primer pairs that were complementary to different regions of the respective mRNAs (Fig EV3E–H). Two negative controls were used: adding $H_2O$ instead of cDNA in the PCR, and cDNA from a control lung tissue was used as template that would give unequivocally negative results. Three different positive controls were used: Serial dilutions ($10^3$ copies to 1 copy) of the plasmids containing the cloned PCR product were used as "calibrator" to determine the linear range of the system; cDNA from human lung cancer cell lines A549 and H520 was used as the "analytical standard" that should give unequivocally positive results and cDNA from human LC biopsies was used as the "biological standard" that provides unequivocally positive results. All PCRs were performed at least in triplicates. The PCR results were normalized with respect to the housekeeping gene *TUBA1A*.

### Isolation and characterization of exosomes from EBCs

Exosomes were isolated from EBCs using ExoQuick-TC Exosome precipitation solution (SBI) with minor changes. EBCs (500 μl) were thawed on ice for 5 min, and 120 μl of ExoQuick-TC was added to the EBC. Exosomes were precipitated by 6 h incubation at 4°C and centrifugation at $1,500 \times g$ for 30 min at 4°C. Exosome pellets were lysed in 350 μl RLT Plus Buffer (RNeasy Micro Kit, Qiagen), and 200 ng of 16S- and 23S-ribosomal Spike-In RNA (Roche) was added to the lysate. Total RNA was isolated using the RNeasy Micro Kit (Qiagen), and isoform-specific expression analysis was performed as explained above.

Total protein extracts from control and LC snap-frozen tissue samples were analyzed by Western blotting following standard protocols (Singh *et al*, 2014) and using antibodies specific for CD63 (ab8219, Abcam), TSG101 (sc-7964, Santa Cruz), and ACTB (ab6276, Abcam).

### Classifier construction and LC score

Log2-transformed Em/Ad ratios of *GATA6* and *NKX2-1* were used as independent variables to predict LC. A linear kernel support vector machine (SVM) (Dimitriadou *et al*, 2010) was used to construct a linear classifier by combining the Em/Ad ratios of *GATA6* and *NKX2-1* of each sample. SVM learning was done with the default parameters, without any adjustments. The SVM finds a robust separating hyperplane and the distance to this hyperplane is our decision score, which we call LC score. The LC score can be conveniently calculated as

$$\text{LC score} = \left(0.715 \times log_2\left(\frac{GATA6\,Em}{GATA6\,Ad}\right)\right.$$
$$\left. + log_2\left(\frac{NKX2-1\,Em}{NKX2-1\,Ad}\right) \times 0.855 + 1.312\right)$$

A sample with LC score >0 will be classified as a lung cancer sample; otherwise, the samples are classified as control samples (Appendix Tables S2 and S3).

The linear SVM was chosen for this study because it is less sensitive to (unbalanced) sample group sizes and/or batch variations, thereby providing robust and reproducible results irrespective of the minor variations that might be present in clinical settings. SVM was also preferred over other common modeling approaches for developing predictors due to several reasons. Linear discriminant analysis (LDA) relies on the assumption of normally distributed data, which does not apply to our data set. K-nearest neighbors (kNN) lead to complex classifier and do not give rise to a score like our LC score. Logistic regression gives also linear decision boundaries, but it is more sensitive to extreme samples/values than SVM. Other strength of our SVM-based approach, when compared to standard expression analysis (Wang & Huang, 2011), is the use of transcript isoform expression ratios of two different genes because it incorporates an additional normalization step to our assay reducing variability coming from biological parameters.

### Statistical analysis

Cell line and mouse experiments were performed three times. Samples were analyzed at least in triplicates. Statistical analysis was performed using R (R Core Team, 2014). In the main figures, the

data are represented as box plots which indicate the first quartile (bottom of the box), the median (line in the middle of the box) and third quartile (top of the box). The whiskers end at the highest and lowest data values. The five-number summaries are given in Tables EV1–EV3. Depending on the data, different tests were performed to determine the statistical significance of the results. The values of these tests are also given in Tables EV1–EV3. Since the Em/Ad expression ratios were not normally distributed according to the Shapiro–Wilk test ($P < 0.01$ for all groups), we performed a Kruskal–Wallis test for the results in Figs 1B and C and 2A–C (Table EV1). The LC score data were normally distributed based on the Shapiro–Wilk test ($P = 0.02375$, Ctrl group; $P = 0.1448$, LC group). Thus, a Tukey's honestly significant difference (HSD) test was performed after one-way analysis of variance (ANOVA) for the results presented in Figs 1D and 3B and C (Table EV2). For Fig 3A, we performed a Tukey's HSD after multivariate analysis of variance (MANOVA; Table EV3). In the EV Figs, the data are represented as mean ± standard error (mean ± s.e.m.). One-way ANOVA was used to determine the levels of difference between the groups. $P$-values < 0.05 were considered as statistically significant.

Two files are provided for reproducing the results presented in the main figures and in the tables: an R Markdown file (GATA6_NKX2_1_EBC.Rmd) together with a file containing the raw data (Raw_data.csv) as Dataset EV1. (i) Both files have to be located into the folder, in which the output of the R Markdown should be saved. (ii) To run the R Markdown file, R version ≥ 2.15.0 and R Studio are required. (iii) After opening the R Markdown file with R Studio, the Knitr option should be selected. (iv) Using the raw data, the R Markdown will generate an html file containing all the plots from the main figures. In addition, several txt files containing the data from Table 2 and Tables EV1–EV4 will be also generated.

### Patent information

Pending patent applications PCT/EP2014/060489 (published as WO 2014/187881), EP 13 16 8629.7, EP 14 00 697.1, EP 14 19 5027.9, US-2016-0244842-A1.

Expanded View for this article is available online.

### Acknowledgements

We thank R. Bender for technical support; K. Müller for support in sample collection; M. Besssler. J. Kwon for reagents; and M. Wheeler for helpful discussions. G. Barreto is funded by the "LOEWE-Initiative der Landesförderung" (III L4–18/15.004 2009) and the DFG grant BA 4036/1-2. A. J. Romero-Olmedo received a doctoral fellowship from CONACyT—COCyT (CVU 510283). This work was done within the Russian Government Program of Competitive Growth of the Kazan Federal University.

### Author contributions

AM, SD, JC, AJR-O, RS, C-MC, IS, EG-D, SGü and GB designed and performed the experiments; AT, GD, JB, SGa, LF, URR, ONI, RHD, SB, WS, TB, and AG were involved in study design; GB, JC, AM, SD, AT, and AG designed the study, analyzed the data, and wrote the manuscript. All authors discussed the results and commented on the manuscript.

### Conflict of interest

The authors declare that they have no conflict of interest.

### The paper explained

#### Problem

Lung cancer is the leading cause of cancer-related deaths worldwide. Despite significant advances in therapeutic strategies, the case fatality rate of lung cancer remains exceptionally high (95%), since the majority of the patients are diagnosed at late stages when patient prognosis is poor. Current diagnostic strategies include chest X-ray, low-dose helical computed tomography, sputum cytology, and invasive sampling. Biochemical and molecular analysis of exhaled breath condensates (EBC) may represent another, non-invasive approach, but has shown inconsistent results in the past.

#### Results

In this study, we have developed an accurate lung cancer diagnostic test based on the expression analysis of embryonic versus adult isoforms of two genes (*GATA6*, *NKX2-1*) in exhaled breath condensates. This diagnosis method is based on specific molecular changes during lung cancer initiation and achieves a high sensitivity of 98.3% and specificity of 89.7%.

#### Impact

We propose this method as an accurate, straightforward, non-invasive and low-price option to achieve LC diagnosis in suspected cases not subjectable to invasive sampling and a valuable addition to LC screening procedures based on low-dose computed tomography.

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
