## [Review Process File · EMBO Molecular Medicine]

Non-invasive lung cancer diagnosis by detection of GATA6 and NKX2-1 isoforms in exhaled breath condensate

Aditi Mehta, Julio Cordero, Stephanie Dobersch, Addi Romero-Olmedo, Rajkumar Savai, Johannes Bodner, Cho-Ming Chao, Ludger Fink, Ernesto Guzman-Diaz, Indrabahadur Singh, Gergana Dobrev, Ulf Rapp, Stefan Günther, Olga Ilinskaya, Saverio Bellusci, Reinhard Dammann, Thomas Braun, Werner Seeger, Stefan Gattenlöhner, Achim Tresch, Andreas Günther, and Guillermo Barreto

Corresponding author: Guillermo Barreto, Max-Planck-Institute for Heart and Lung Research

Review timeline:

Submission date:	08 March 2016
Editorial Decision:	02 May 2016
Revision received:	03 August 2016
Editorial Decision:	12 September 2016
Revision received:	23 September 2016
Accepted:	28 September 2016

Transaction Report:

Editor: Roberto Buccione

1st Editorial Decision

02 May 2016

Thank you for the submission of your manuscript to EMBO Molecular Medicine. We have now heard back from two Reviewers whom we asked to evaluate your manuscript.

We are very sorry that it has taken so long to get back to you on your manuscript. In fact, we experienced unusual difficulties in securing three willing and appropriate reviewers and in obtaining their evaluations in a timely manner and finally, we were also unable to further contact a reviewer (#2), and for latter reason I am now proceeding based on the two available, consistent evaluations. I trust that the inevitable frustration due to the unusual delay will be somewhat tempered by the fact that the Reviewers are quite supportive and, in my opinion, offer valuable suggestions to improve your manuscript.

I will not go into much detail as the comments are exhaustive and clear in my opinion, and should not prove too challenging to address.

You will see that the Reviewers raise a number of issues including the excess "optimism" presented with respect to predictive power/clinical relevance, inadequate consideration of potential confounders, and other technical issues. It is for instance noted that you often say and imply that the test can be used for "early LC diagnosis" but an important caveat is that all of the lung tumors included in the study were clinically apparent ones. The other reviewer would like you to better explain, among other things, why consider EBC and not blood samples. Reviewer 1 also especially appreciated the provision of the checklists and the R script, but suggests several improvements.

I should add that, during our cross-commenting procedure, both reviewers agreed that the study is valuable but needed improvement. It was also noted that the R script could be made more user friendly so that one could easily identify the appropriate input data and run the script with little modification other than changing directory name.

Finally, I concur with reviewer 3 that the manuscript could do with some extensive re-working of the language and presentation to make reading easier and more accessible to a broader readership.

In conclusion, while publication of the paper cannot be considered at this stage, given the potential interest of your findings, we would be pleased to consider a revised submission, with the understanding that the Reviewers' concerns must be addressed with additional experimental data where appropriate and that acceptance of the manuscript will entail a second round of review.

Please note that it is EMBO Molecular Medicine policy to allow a single round of revision only and that, therefore, acceptance or rejection of the manuscript will depend on the completeness of your responses included in the next, final version of the manuscript.

As you know, EMBO Molecular Medicine has a "scooping protection" policy, whereby similar findings that are published by others during review or revision are not a criterion for rejection. I understand that in this case, to address the above might entail a significant amount of additional work and time and might be technically challenging. However, I do ask you to get in touch with us after three months if you have not completed your revision, to update us on the status. Please also contact us as soon as possible if similar work is published elsewhere.

EMBO Molecular Medicine now requires a complete author checklist (<http://embomolmed.embopress.org/authorguide#editorial3>) to be submitted with all revised manuscripts. I am attaching a copy of the checklist to this letter for your convenience.

Please note that we now mandate that all corresponding authors list an ORCID digital identifier. You may do so through our web platform upon submission and the procedure takes <90 seconds to complete. We also encourage co-authors to supply an ORCID identifier, which will be linked to their name for unambiguous name identification.

Last, but not least, please carefully conform to our author guidelines (<http://embomolmed.embopress.org/authorguide>) to ensure rapid pre-acceptance processing in case of a favourable outcome on your revision.

I look forward to seeing a revised form of your manuscript as soon as possible.

***** Reviewer's comments *****

Referee #1 (Remarks):

This manuscript reports a linear score constructed from normalized ratios of expression levels of isoforms of GATA6 and NHX2-1 in exhaled condensate for lung cancer diagnosis ("LC-score"). Overall the paper is clearly written and provides a thorough account of the development and validation of the LC-score. The authors provide completed STARD and REMARK reporting checklists, and they explain how they have addressed the relevant aspects of the NCI Omics Checklist. An R script and raw data are also provided to allow interested readers to reproduce the calculations.

Major comments

1. An important caveat for interpretation of the results is that all of the lung tumors included in the study were tumors that were clinically apparent. The authors need to be more cautious in the interpreting the evidence for clinical usefulness of the test. The phrase used repeatedly in the manuscript is that the test can be used for "early LC diagnosis." While the test appears to have some promise because it seems able to detect early stage tumors as well as more advanced stage tumors,

that is not the same as demonstrating that the test is able to noninvasively detect tumors before they are clinically apparent. The study is not designed to determine whether the test can detect subclinical disease, and that would require a very large prospective screening study. Yet another use of the test could be to triage cases with suspicious image findings for follow-up biopsy, but that might require a different study design as well. The authors need to temper their conclusions about the clinical use of the test in the abstract and add an explanation in the Discussion section describing how they propose that the test could be used and what are the limitations of their study design for supporting such use.

2. Related to comment #1, if this test was really used clinically, the performance metrics of positive and negative predictive value would be very important. The authors should discuss the population to which they would apply this test and use estimates of LC prevalence appropriate to the intended use population to calculate PPV and NPV. Similarly, overall accuracy would also be affected by prevalence. If detection of subclinical disease is what the authors have in mind, then sensitivity and specificity might also be diminished due to possible spectrum bias.

3. From Table S1 it appears that more than approximately 20% of the patients with lung cancer had either recurrent disease or had already initiated treatment by the time the EBC sample was collected. These could be important confounding factors and the authors should perform a sensitivity analysis in which they present estimates of the test performance excluding these cases.

4. The authors should discuss whether there were any factors that appeared to be associated with incorrect classifications. For example, did the incorrectly classified cases tend to be those with LC-score near zero, lower stage disease, shorter duration of EBC sample collection, etc?

Other comments

5. My impression is that the authors narrowed their focus to the two genes GATA6 and NHX2-1 prior to examining the EBC data and building the score, but they should confirm whether this is a correct understanding. The strength of the evidence would be viewed differently if these two genes were selected only after some preliminary analysis of a larger number of genes measured on the EBC samples.

6. There is reference made to "Table I" in several places, but I could not find this table in the materials made available to the reviewers.

7. Main manuscript, page 14: It is stated that there are four different transcripts of NKX2-1 but in the next sentence there is reference to "both" isoforms of NKX2-1. Please clarify.

8. Main manuscript, page 16: The comparison of sensitivity and specificity of CT to the EBC test is not really fair as the screening setting is likely to have a different spectrum of disease than the setting in this study which used LC cases that already had clinically apparent disease.

9. Several of the figures presented were either barcharts (e.g., Figure S10) or displayed dotplots but annotated them with horizontal bars for the mean and SEM. It would be preferable to use displays such as boxplots with individual points displayed (jittering as needed) or violin plots. Mean and SEM could be given in the figure legends as needed.

10. Main manuscript, page 6, line 11: Insert "analysis" between "expression" and "of GATA6."

11. Supplementary Results, page 1: Do the authors have any explanation for the reversal of the gene expression levels in cell lines relative to lung tissue?

12. Supplementary Material & Methods, page 34: Did EBC collection time differ between LC cases and controls? Was collection time related to gene expression levels?

13. Supplementary Material & Methods, page 37: The formula presented here for the LC score differs from the one on page 11 of the main manuscript and in response to item 6 of the NCI omics checklist. Is one of the versions developed on the training data only and the other the "improved version" that was re-fit on the combined training and test data?

14. Supplementary Material & Methods, page 38: The description of the ANOVA test could be improved to say something like "ANOVA was used to test whether there were differences in the mean levels . . ."

15. NCI Omics checklist, item 8: I believe that the intent of this item is to assess analytical performance of the assay and reported score rather than the clinical performance. It would be helpful to know the repeatability and reproducibility of the LC scores assessed on replicate samples from the same study participant, for example by display of Bland-Altman plots.

16. STARD checklist, item 13: The response given is "NA" but see comment #15.

17. STARD checklist, item 17: The response given is "NA" but it does seem relevant as some portion of patients had received treatment prior to collection of EBC sample.

18. STARD checklist, item 18: The response given is "NA" but lung cancer stage, COPD, and IPF are all relevant.

19. If the completed STARD and REMARK checklists will be provided with the published paper, a better mechanism for indicating page number will be required as page numbering will change and it will be necessary to indicate whether the numbering refers to the main paper or one of the several supplements.

20. I did not attempt to run the R script on the data file provided in the Excel file, but it did not appear that the data were the same as indicated in the R script as the number of rows were different.

Referee #3 (Remarks):

I like to congratulate the authors with their work and encourage them to proceed this research. It is an important upcoming research field where non-invasive diagnosis for lung cancer is being explored. Nevertheless, I have some comments and remarks about the manuscript.

General comments

- Please provide conflict of interest if there is any.
- In the significance section, it is stated that this method complements the success of CT: did you add the EBC results to the outcome of a CT scan in order to identify potential nodules as being malignant and where in the process would you place the EBC test: before or after the LDCT scan?
- Why do it in exhaled breath condensate and not in blood samples? It takes longer time to obtain exhaled breath condensate and are the methods reliable?
- Also, were all the patients able to give exhaled breath? Since the collection of exhaled breath condensate takes some time, patients need to sit with a nose-clip and breath through a mouthpiece. We see in the clinic that patients which are restricted have some trouble to perform this manoeuvre or experience this as oppressive.
- Did you take into account the recommendations for EBC sampling by the ERS/ATS task force (Horvath et al., European Respiratory Journal 2005; 26:523-548, DOI:10.1183/09031936.05.00029705)?
- When reading the main article file for the first time, it felt overwhelming and rather confusing. A lot was explained in supplement. However, if possible, please restructure the text so that the main article file can be used as stand-alone article without the need for a supplement and in that way, supplement can contain extra figures or materials and methods, but important results should be included in the main manuscript. I felt 'lost' between main files, supplementary results and methods and figures.

Introduction (page 5)

- It seems the reference from the Globocan website is missing. On the website is an example on how to refer to this website. Please provide a reference, with the date of consultation.
- Please provide a reference for the following sentence: "Approximately 55-85% of the CT-detected LC can be surgically removed resulting in high five-year patient survival of almost 52%."

- Is there any evidence for your postulation that many of the mechanisms involved in embryonic development are recapitulated during LC initiation. If so, please provide a reference.
- What made you choose for GATA6 and NKX2-1 and not the other regulators in lung development as reported by Maeda et al., 2007?
- In the last sentence, the results are already mentioned, which I feel should not be done here. Please rephrase your postulation so that it becomes a clear hypothesis.

Results

- In the first phase, you optimized the SOP for RNA isolation out of FFPE samples. The second phase was the optimization of a SOP for EBC collection and storage, but then you used the SOP for RNA isolation out of FFPE material for the isolation out of EBC. Shouldn't an optimization be needed when collecting RNA from EBC or was the SOP exactly the same?
- It seems table 1 is missing
- In table S1, it seems that the number of current smokers have doubled in the control group of the validation set compared to in the training set (20.3% vs. 9.2%). Could this have influenced the results? Is a Chi²-test used to look for significant differences between the groups?
- Controls? How were they chosen, were they followed prospectively and for how long? Was only EBC obtained? How could you be sure the healthy controls were healthy? How did you define COPD and IPF. Were there differences between smokers and non-smokers in the lung cancer group or between different stages of COPD, lung cancer? Were there differences between the lung cancer subtypes? Were there patients with COPD and lung cancer? Is there an influence of smoking (are there significant differences when comparing smokers to non-smokers)?
- Furthermore, it seems that the number of lung cancer patients overwhelms the numbers of controls, which could induce selection bias and hence, overoptimistic results. I suggest to include more controls in future work.
- Why standardize to TUBA1A, while in supplement the genes GAPDH or HPRT1 are used? Wasn't the same SOP used?
- Are the Em/Ad ratio's reported as mean or median? Furthermore, the numbers mentioned (ratio's for controls for GATA6 and NKX2-1 of 0.256 and 0.186 and for lung cancer patients 1.792 and 2.228) seem not to be in accordance to figure 1B.
- Figure S2: how did you obtain healthy lung tissue? From the same patients (were the patients their own control), or from patients undergoing biopsies for other lung diseases (and are not "healthy")? This is mentioned in supplement but I think should be placed in the original manuscript file.
- Can you explain why Em/Ad ratio's dropped for GATA6 in stage III lung cancer patients and why this is not the case for NKX2-1?
- Why isn't a plot shown after grouping by smoking status in Figure 1C?
- How come there are no stage IV lung cancer patients in figure 1D when they are mentioned in table S1? Furthermore, why are only NSCLC patients shown and not SCLC?
- Fig 2B: are there any explanations why there are some significant differences in patients between tissue and EBC ratio's?
- Figure S7C: the ROC curve shows very optimistic accuracy for the LC score. However, is the LC score the combination of both GATA6 and NKX2-1 scores or the SVM outcome? What exactly is meant with "point of operation"? Shouldn't this be the most upper left point?
- Figure S7: why are the ratio's log₂-transformed and not log₁₀-transformed like in figure S5?
- Page 11: the external validation was done on 68 controls. Are those only healthy controls or also include COPD/IPF controls? Please clarify in the text.
- Figure S8A-C: it is stated that the figure includes both the training and validation set. However, the training set should not be included since the validation set is independent. Same remark for figure S8-D. A training set is used to train a certain method and should not be used for validation. This will result in overly optimistic outcomes. If you want to have a larger number, you should include more patients in the validation set instead of combining validation and training set.
- How come that in figure 3, stage IV lung cancer is included and not in figure 1D?
- In the figure caption of figure 3C, it is stated that data are represented as in figure 2B, to me this does not make any sense, since this is a totally different figure.
- Figure 3D shows the separation of IPF, healthy controls and lung cancer patients. However, in figure S8 it is not shown to be able to discriminate IPF. How was the training done to discriminate IPF? Is the same thing done for COPD patients?
- Colour code in Figure S9 is not correct.
- In supplement material and methods it is mentioned that patients were sequentially included first in Mexico, then in Waldhof Elgershausen and finally in Giessen and Marburg. Because samples were

not taken in parallel but sequentially, this could introduce bias due to differences location. Are the results robust enough and remain the same when a correction for location is done?

- Further in supplement material and methods, the SOP for EBC collection is described. However, since this is an important factor in the study, I would suggest to include a part of this in the manuscript file. Furthermore, it is stated that every donor performed 10 minutes of tidal breathing but if this was too long, 5 minutes were also OK without loss in quality. Next to this, it is stated in the OMICs guidelines that EBC samples obtained in a time <5min were discarded and 10 minutes was optimal. However, when specifying the minimum amount of specimen needed, 5 min delivers approximately 500µl which is enough. So why wasn't 5 minutes used then as reference instead of the 10 minutes, which is still 5 minutes less for the patient?
- In the OMICs guidelines (point 8) it is stated that an independent set of 79 previously unseen samples are used for external validation. However, in the text it states that 138 previous samples are used. Why this discordance?

Discussion

- How do you see the future? The test as it is described now, it feels like a stand-alone diagnostic test. Can it be used that way? Furthermore, the last paragraph is attributed to the comparison with CT scans. But where would you put the test in the diagnostic work-up? And what should happen when the breath test is positive?

Materials and Methods

- Was the study performed in accordance with the declaration of Helsinki? If so, please provide this in the study design section.
- In figure S1, in the start-1 phase you start with n=117. After excluding 5, you obtain n=151. How is this possible? Please provide more details on the number of samples.
- It seems table 1 is missing
- The reference to "R" is not correct.
- What are the inclusion/exclusion criteria?
- Please give more information about how the sample was taken, stored and shipped.
- What p-value was considered to be the threshold for statistical significance?
- The pages in the STARD checklist seem not to correspond and a lot of this important data is implemented in the supplement, which I think should be done in the main manuscript file.
- Following STARD checklist, no intervals for sensitivity, specificity etc. are given.
- In the REMARK checklist, a figure 4 is mentioned. However, through the manuscript, no figure 4 was shown or discussed. Please update the checklists.

1st Revision - authors' response

03 August 2016

Please find as an attachment our revised manuscript (EMM-2016-06382) with the title "Detection of *GATA6* and *NKX2-1* isoforms in exhaled breath condensate for lung cancer diagnosis", which we would like to resubmit as a Report to *EMBO Molecular Medicine*. As you will confirm, we have addressed ALL concerns raised by the Reviewers. Their constructive and stimulating suggestions guided us to substantially improve our manuscript (ms). We would also like to express our gratitude for your support and editorial work. You will find appended to this letter, a point-by-point response to all the concerns of the Reviewers. Nevertheless, I will summarize in the following five points the major improvements in our ms using as the headline for each point text that you mentioned in the decision letter:

A. "...*Extensive re-working of the language and presentation to make reading easier and more accessible to a broader readership...*" I decided to start with this point to emphasize the changes done to the overall structure of the ms. Following your advices and the ones from both Reviewers, we have changed the overall structure of the ms, simplifying it and making easier to read as stand-alone article avoiding switching to the Appendix Supplementary Materials. Important information for the main idea of the article have been included into the main text, whereas information for specialists that required a deeper view into the data were included into the Expanded View Figures and Tables or in the Appendix Supplementary Materials.

B. "...*the excess "optimism" presented with respect to predictive power/clinical relevance...*" Following the suggestions from both Reviewers, we have changed the text of the ms, being more

cautious in our interpretation of the evidence for clinical usefulness of our LC diagnosis method. In the revised version of our ms, we do not claim any more that our method can be used as stand-alone assay for early LC diagnosis. Instead, we mention in the Results section that our method is able to detect early staged samples implying that they were clinically apparent at the time of analysis. Nevertheless, we emphasized that our method has a great potential, since it was externally validated on an independent set of EBCs AND it is able to detect LC at stages I and II.

In addition, we would like to refer you to the Discussion section in our ms, where we comment on the limitations of our study, as well as we propose how to use our method:

“...Even though our LC diagnosis method was externally validated on an independent set of EBCs, the results of our study are insufficient to safely predict its usefulness under clinical conditions, for which a suitably designed, large prospective study would be required. Despite the limitations of our study, the results presented here are promising, since our method is able to detect LC at stages I and II. We propose to incorporate our method into the current protocols for patients undergoing diagnostic evaluation for pulmonary diseases characterized by hyperproliferation...”

C. “...*Inadequate consideration of potential confounders, and other technical issues...*” In the improved version of the ms, we have addressed these potential confounders and technical issues. As compared to the previous version, the resubmitted ms provides a vastly improved statistical analysis of the data as well as new results that strengthen and support our claims:

C.1 The plots in the main figures are now box plots, as requested by Reviewer 1 (see point 9 in the response to the Reviewers). In addition, we have elaborated reproducible statistical analysis of our data using R. The methods used for statistical analysis are described in the Material and Methods section. The five-number summaries and the statistical test values are summarized in the newly added Tables EV1-EV3.

C.2 We have excluded the reiteration of the SVM-based classifier using the training and the validation sets of EBCs together. Reviewer 3 requested to keep both sets of samples separated (see point 27 in the response to the Reviewers). With this exclusion, the ms become easier to read and to understand, without any loss in the impact of the major findings.

C.3 The control population in the different sets of samples contains IPF and COPD samples. We have explained the rationale for this reassignment in the description of the study population in the Materials and Methods Section as follows:

“...The control population for the analysis of FFPE samples included lung tissue that was taken from macroscopically healthy adjacent regions of the lung of LC patients and control lung tissue that was obtained from age-matched donor lungs, who have had no diagnosis or family history of LC, in the frame of surgical size reduction of the donor lung during lung transplantation. Lung tissue samples from IPF (idiopathic pulmonary fibrosis) and COPD (chronic obstructive pulmonary disease) patients were also included in to the control population because these lung diseases have been reported to increase LC risk when compared to individuals with normal pulmonary function (Li et al, 2014; Turner et al, 2007). In addition, the IPF and COPD cohorts were included in the study to determine the discriminatory power of the diagnosis method proposed here with respect to other non-cancer diseases characterized by alveolar or bronchiolar hyperproliferation...”

Importantly, the control population was representative of the LC population with respect to age, smoking history as well as gender distribution (Table 2).

Further, we would like to attract your attention to our comments in the Discussion section supporting the reassignment of the control population and arguing against potential bias:

“...As described in the study population, IPF (idiopathic pulmonary fibrosis) and COPD (chronic obstructive pulmonary disease) samples were included into the control groups since they are non-malignant hyperproliferative lung diseases with an increased risk of LC (Li et al, 2014; Turner et al, 2007). Moreover, IPF and COPD are frequently found comorbidities in LC. Consistent with these findings, four of the eight wrongly classified control samples in the validation set were COPD samples (Appendix Table S3). Further, performance assessment values of the LC score using a control population with (Fig. 2E) or without IPF and COPD samples (Table EV4) were similar

arguing against a potential bias by incorporating the IPF and COPD samples into the control population...”

C.4 We have included into the revised ms two tables with performance metrics (Fig 2E and Table EV4), following the suggestions from both Reviewers (see points 2; 3 and 14 in the response to the Reviewers). These tables contains the positive predictive value (PPV), negative predictive value (NPV), true positive rate (TPR, sensitivity) and true negative rate (TNR, specificity) of the LC score when applied either to the validation set of EBCs (Fig 2E) or to sub-groups of our samples (Table EV4), for example excluding nonsmokers, or excluding patients with recurrent diseases, etc. In all the sub-groups the LC score achieved high performance values correlating with the values obtained after the external validation and arguing against potential bias related to sample distribution.

Furthermore, we followed the suggestion of the Reviewer 1 and calculated the performance of the LC score when applied to high risk groups, such as defined by (Bach et al, 2003) for smokers having at least 20 pack-years of smoking exposure. We would like to refer you to the newly added Appendix Table S4 and the corresponding text in the Appendix Supplementary Results with the title “Estimation of the LC score performance metrics for smoking-related lung cancer”.

C.5 The repeatability of our method was determined in the newly added Fig EV4B by Bland-Altman plots as suggested by the Reviewer 1 (see point 15 in the response to the Reviewers). Due to space limitation of a Report at EMBO Mol Med, we briefly mention in the Results section of the main ms:

“...The repeatability of isoform-specific expression analysis of *GATA6* and *NKX2-1* was confirmed by Bland-Altman plots (Bland & Altman, 1986) after measurements in two EBCs from the same patient, in five patients (Fig EV4B, Appendix Supplementary Results)...”

However, we would like to refer you to the text describing in detail the Fig EV4B in the Appendix Supplementary Results.

C.6 Fig EV1A contains new data showing the complementary and developmentally regulated expression of Em- and Ad-isoforms of *Gata6* and *Nkx2-1* during mouse lung development.

C.7 The newly added Figs EV1D-E demonstrated that RNA-containing exosomes are enriched in EBCs of LC patients, thereby providing a plausible explanation for the presence of the *GATA6* and *NKX2-1 transcripts* in the EBCs.

D. “...*The other reviewer would like you to better explain, among other things, why consider EBC and not blood samples...*” A explained to the Reviewer 3 (see point 3 in the response to the Reviewers), one of the main reasons to initiate this project using EBCs was the fact that we wanted to detect primarily lung diseases, thereby being EBCs closer correlated than blood samples. An additional reason is the future development of our method of LC detection using EBCs, which involves more advanced technology as the electronic Nose (eNose), which is a highly sensitive analysis platform for detection of diseases. Some of or co-authors are working in that direction. Nevertheless, we would like to clarify that we are currently following this valuable suggestion from Reviewer 3. We initiated blood sample collection and measurements in parallel to EBCs analysis. However, these data will be the scope of future work.

E. “...It was also noted that the R script could be made more user friendly so that one could easily identify the appropriate input data and run the script with little modification other than changing directory name...” We also agree with this suggestion from the Reviewers. In order to facilitate easier understanding and the reproduction of our results using the R script, we have provided two additional files and included in the Materials and Methods section of the main ms the following description of how to use them:

“...Two files have been submitted for reproducing the results presented in the main figures and in the tables: an R Markdown file (*GATA6_NKX2_1_EBC.Rmd*) and a file containing the raw data (*LC_Data.csv*). (I) Both files have to be located into the folder, in which the output of the R Markdown should be saved. To run the R Markdown file, R version $\geq 2.15.0$ and R Studio are required. (II) After opening the R Markdown file with R Studio, the Knitr option should be selected.

(IV) Using the raw data, the R Markdown will generate an html file containing all the plots from the main figures. In addition, several txt files containing the data from Tables 2, EV1-4, will be also generated..."

Since you previously commented that it was difficult to find Reviewers for our ms, we would like to take the liberty to suggest the following lung cancer specialists as potential Reviewers, in case that you faced again similar difficulties:

Georgios T. Stathopoulos, MD PhD. Associate Professor, Department of Physiology, and Principal Investigator, Laboratory for Molecular Respiratory Carcinogenesis. Faculty of Medicine, University of Patras. 26504 Rio, Greece. Tel: +30-2610-969154. Email : gstathop@upatras.gr

Thomas R. Muley, Ph.D. Head of Biobank and Tumor Documentation at the Thoraxklinik at Heidelberg University Hospital. 69126 Heidelberg. Germany. Tel: +49 (0)6221 396-1110. Email: thomas.muley@thoraxklinik-heidelberg.de and orthomas.muley@med.uni-heidelberg.de

We strongly believe that our work constitutes a significant contribution to the general scientific community. *EMBO Molecular Medicine* will be the best platform to make our findings accessible to a broad audience. We hope that the Editorial Staff of your recognized journal will confirm its positive view about the novelty and quality of our work and will grant the final acceptance for publication of our manuscript as a Report in your renowned journal. We appreciate the time and effort that the Editorial Staff spent evaluating our manuscript. We deeply appreciate the constructive editorial work from *EMBO Molecular Medicine*.

Point-by-point response

Reviewer #1 (Remarks):

This manuscript reports a linear score constructed from normalized ratios of expression levels of isoforms of GATA6 and NHX2-1 in exhaled condensate for lung cancer diagnosis ("LCscore"). Overall the paper is clearly written and provides a thorough account of the development and validation of the LC-score. The authors provide completed STARD and REMARK reporting checklists, and they explain how they have addressed the relevant aspects of the NCI Omics Checklist. An R script and raw data are also provided to allow interested readers to reproduce the calculations.

We would like to thank Reviewer 1 for the time and effort expended during the review of our manuscript. We also appreciate the positive comments from Reviewer 1.

Major comments

1. An important caveat for interpretation of the results is that all of the lung tumors included in the study were tumors that were clinically apparent. The authors need to be more cautious in the interpreting the evidence for clinical usefulness of the test. The phrase used repeatedly in the manuscript is that the test can be used for "early LC diagnosis." While the test appears to have some promise because it seems able to detect early stage tumors as well as more advanced stage tumors, that is not the same as demonstrating that the test is able to noninvasively detect tumors before they are clinically apparent. The study is not designed to determine whether the test can detect subclinical disease, and that would require a very large prospective screening study. Yet another use of the test could be to triage cases with suspicious image findings for follow-up biopsy, but that might require a different study design as well. The authors need to temper their conclusions about the clinical use of the test in the abstract and add an explanation in the Discussion section describing how they propose that the test could be used and what are the limitations of their study design for supporting such use.

We completely agree with this point from Reviewer 1. Following the suggestions, we have changed the text of the manuscript (ms), being more cautious in our interpretation of the evidence for clinical usefulness of our LC diagnosis method. In the revised version of our ms, we do not claim any more that our method can be used as stand-alone assay for early LC diagnosis. Instead, we mention in the Results section that our method is able to detect early staged samples implying that they were clinically apparent at the time of analysis. In addition, we comment in the Discussion section the

limitations of our study. We also propose how to use our method as follows:

“...Even though our LC diagnosis method was externally validated on an independent set of EBCs, the results of our study are insufficient to safely predict its usefulness under clinical conditions, for which a suitably designed, large prospective study would be required. Despite the limitations of our study, the results presented here are promising, since our method is able to detect LC at stages I and II. We propose to incorporate our method into the current protocols for patients undergoing diagnostic evaluation for pulmonary diseases characterized by hyperproliferation. LC score might be also used to improve the sensitivity of risk prediction models for LC as previously suggested for biomarkers (Spitz et al, 2008). In addition, we suggest to integrate our technology into CT-based LC screening approaches in high risk populations (Bach et al, 2003; Cassidy et al, 2008; Colditz et al, 2000; de Torres et al, 2007; Spitz et al, 2007), a procedure routinely used in the US, but not in Europe due to concerns regarding the very high percentage of false positive observations (>90%) and hence low specificity (73.4%) (National Lung Screening Trial Research et al, 2013), resulting in unnecessary follow-up CT scans, bronchoscopy or even surgery (Jett, 2005). Concomitant implementation of EBC-based LC detection together with CT could help to reduce the false-positive rate of CT imaging, e.g. in cases with suspicious image findings, thereby preventing individuals from being unnecessarily exposed to high dose of radiation or surgery....”

2. Related to comment #1, if this test was really used clinically, the performance metrics of positive and negative predictive value would be very important. The authors should discuss the population to which they would apply this test and use estimates of LC prevalence appropriate to the intended use population to calculate PPV and NPV. Similarly, overall accuracy would also be affected by prevalence. If detection of subclinical disease is what the authors have in mind, then sensitivity and specificity might also be diminished due to possible spectrum bias.

We would like to thanks for this constructive comment. Reflecting the importance of the information suggested by Reviewer 1, we have incorporated into the main ms two tables (Fig 2E and Table EV4). These tables contains the positive predictive value (PPV), negative predictive value (NPV), true positive rate (TPR, sensitivity) and true negative rate (TNR, specificity) of the LC score when applied either to the validation set of EBCs (Fig 2E) or to sub-groups of our samples (Table EV4). The performance of the LC score in the sub-groups tested was high correlating with the performance during the external validation.

Furthermore, we followed the suggestion of the Reviewer 1 and calculated the performance of the LC score when applied to high risk groups, such as defined by (Bach et al, 2003) for smokers having at least 20 pack-years of smoking exposure. We would like to refer Reviewer 1 to the Appendix Table S4 and the corresponding text in the Appendix Supplementary Results with the title “*Estimation of the LC score performance metrics for smoking-related lung cancer*”:

“...During the external validation of our EBC-based LC diagnosis method, the performance was assessed on a population with a LC prevalence of 43.5% (Table 1, validation set of EBCs) leading to a positive predictive value (PPV) of 0.881 (Fig 2E). If we would estimate the performance metric of our LC diagnosis method on a risk group in a specific population, such as defined by (Bach et al, 2003) for smokers having at least 20 pack-years of smoking exposure, we will have to use the LC prevalence of 7%, specified for this population. Following this rationale, we have estimated the performance metric of the LC score (Appendix Table S4) when applied to a hypothetical population of 100,000 current smokers, using a LC prevalence of 7% (Bach et al, 2003), meaning that 7,000 individuals will develop LC. Since our LC score has a sensitivity of 98.3% (Fig 2E), we will detect 6,881 smokers as true positive and 119 smokers as false negative. On the other hand, there will be 93,000 smokers that will not develop LC. Based on a specificity of 89.7% (Fig 2E), we will detect 83,421 smokers as true negative and 9,579 smokers as false positives. Using these numbers, we obtain a PPV of 41.8% and a NPV of 99.9%. Although we used the sensitivity and specificity determined during the external validation of the LC score, we obtained a much lower PPV. Our observations are in accordance with previous reports (Parikh et al, 2008), the PPV is highly dependent on the prevalence of the disease....”

As stated in the text, Reviewer 1 is right; the PPV is dependent on the prevalence on the disease.

3. From Table S1 it appears that more than approximately 20% of the patients with lung cancer had

either recurrent disease or had already initiated treatment by the time the EBC sample was collected. These could be important confounding factors and the authors should perform a sensitivity analysis in which they present estimates of the test performance excluding these cases.

We would like to refer Reviewer 1 to the answer of the previous point. As mentioned there, we have incorporated into the main ms two tables (Fig 2E and Table EV4). These tables contains the performance metrics of the LC score when applied either to the validation set of EBCs (Fig 2E) or to sub-groups of our samples (Table EV4), for example excluding nonsmokers, or excluding patients with recurrent diseases, etc. In all the sub-groups the LC score achieved high performance values correlating with the values obtained after the external validation.

4. The authors should discuss whether there were any factors that appeared to be associated with incorrect classifications. For example, did the incorrectly classified cases tend to be those with LC-score near zero, lower stage disease, shorter duration of EBC sample collection, etc?

This is an exceptional observation from Reviewer 1, because on a deeper analysis following the Reviewer's advices, we figured out that the wrongly classified control samples during the external validation of the LC score were indeed samples from COPD patients. We have included into the Discussion section the following text:

"...IPF (idiopathic pulmonary fibrosis) and COPD (chronic obstructive pulmonary disease) samples were included into the control groups since they are non-malignant hyperproliferative lung diseases with an increased risk of LC (Li et al, 2014; Turner et al, 2007). Moreover, IPF and COPD are frequently found comorbidities in LC. Consistent with these findings, four of the eight wrongly classified control samples in the validation set were COPD samples (Appendix Table S3)..."

Other comments

5. My impression is that the authors narrowed their focus to the two genes *GATA6* and *NKX2-1* prior to examining the EBC data and building the score, but they should confirm whether this is a correct understanding. The strength of the evidence would be viewed differently if these two genes were selected only after some preliminary analysis of a larger number of genes measured on the EBC samples.

GATA6 and *NKX2-1* were selected for our study due to their similar gene structure as well as their involvement in lung cancer. A sentence explaining this specific point was included into the Introduction section:

"...Among other lung development relevant genes, *GATA6* and *NKX2-1* were selected due to their similar gene structure (Fig 1A) and their implication in LC (Cheung et al, 2013; Winslow et al, 2011)..."

Nevertheless, as Reviewer 1 points out, we did analyze the expression of other genes in both sets of EBCs (training and validation set). For example, *FOXA2* and *ID2*, two lung development relevant genes with similar gene structure as *GATA6* and *NKX2-1*, were analyzed at very early stages of the project. Unfortunately, the expression analysis of these genes was not consistent due to the very low abundance and/or poor stability of their transcript isoforms. Thus we decided to focus on the optimization of expression analysis of *GATA6* and *NKX2-1*.

6. There is reference made to "Table I" in several places, but I could not find this table in the materials made available to the reviewers.

We offer our apologies for this mistake. We have included Table 1 in the revised version of the ms.

7. Main manuscript, page 14: It is stated that there are four different transcripts of *NKX2-1* but in the next sentence there is reference to "both" isoforms of *NKX2-1*. Please clarify.

The text has been changed to:

"...Interestingly, a sequence search in all public mRNA data bases using AceView (<http://www.ncbi.nlm.nih.gov/IEB/Research/Acembly/>) revealed four different transcripts of *NKX2-*

I and two of GATA6, including the Em-and Ad-isoforms reported here...."
We hope that this passage is now easier to understand.

8. Main manuscript, page 16: The comparison of sensitivity and specificity of CT to the EBC test is not really fair as the screening setting is likely to have a different spectrum of disease than the setting in this study which used LC cases that already had clinically apparent disease.

We agree with this comment from Reviewer 1. We have removed this comparison from the revised version of the ms.

9. Several of the figures presented were either barcharts (e.g., Figure S10) or displayed dotplots but annotated them with horizontal bars for the mean and SEM. It would be preferable to use displays such as boxplots with individual points displayed (jittering as needed) or violin plots. Mean and SEM could be given in the figure legends as needed.

We would like to thank Reviewer 1 for this suggestion that helped us to significantly improve our manuscript. All the charts in the main Figures are now presented as box plots. The five-number summary of each box plot and the values of the statistical significance tests are presented in the newly added Tables EV1-3.

10. Main manuscript, page 6, line 11: Insert "analysis" between "expression" and "of GATA6."

The text has been changed accordingly.

11. Supplementary Results, page 1: Do the authors have any explanation for the reversal of the gene expression levels in cell lines relative to lung tissue?

As mentioned in the Appendix Supplementary Results, all the cell lines tested were human cancer cell lines. We interpreted the increased expression of the embryonic isoforms of *GATA6* and *NKX2-1* in these cell lines as indication for the potential use of these transcripts as biomarkers for LC detection. In order to understand the mechanism of transcriptional regulation mediating the switch between both isoforms, we also analyzed the chromatin structure at the promoters of *GATA6* and *NKX2-1* by chromatin immunoprecipitation (ChIP) and the DNA methylation levels by combined bisulfite restriction digestion assay (COBRA) and methylated DNA immunoprecipitation (meDIP) in these cell lines. Our chromatin analysis correlates with the expression analysis of the isoforms. However, these data are the scope of an upcoming manuscript from our group focusing on the mechanism of the isoform switch.

12. Supplementary Material & Methods, page 34: Did EBC collection time differ between LC cases and controls? Was collection time related to gene expression levels?

Collection time between LC and controls was in the majority of the samples 10 minutes. Only if the patient was not able to breath through the collection device for 10 min, we collected the EBC for a minimum of 5 min. The gene expression level did not correlate with the time of collection. There was no significant difference in collection time between LC and controls

13. Supplementary Material & Methods, page 37: The formula presented here for the LC score differs from the one on page 11 of the main manuscript and in response to item 6 of the NCI omics checklist. Is one of the versions developed on the training data only and the other the "improved version" that was re-fit on the combined training and test data?

We solved this issue in the revised version of the ms presenting only one formula in the Material and Methods section. Following the advices from the other Reviewer (Reviewer 3), we have excluded the improvement of the LC score based classifier by using the training and validation set together.

14. Supplementary Material & Methods, page 38: The description of the ANOVA test could be improved to say something like "ANOVA was used to test whether there were differences in the mean levels..."

Following the suggestion from Reviewer 1, we improved the description of the Statistical analysis in the Material and Methods section as follow:

“...Statistical analysis was performed using R (R_Core_Team, 2014). In the main figures, the data are represented as box plots and the five-number summaries are given in the Tables EV1

3. Depending on the data, different tests were performed to determine the statistical significance of the results. The values of these tests are also given in in the Tables EV1-3. Since the Em/Ad expression ratios were not normally distributed according to the Shapiro-Wilk test ($P < 0.01$ for all groups), we performed a Kruskal-Wallis test for the results in the figures 1B-C and 2A-C (Table EV1). The LC score data were normally distributed based on the Shapiro-Wilk test ($P = 0.02375$, Ctrl group; $P = 0.1448$, LC group). Thus, a Tukey's Honestly Significant Difference (HSD) test was performed after one-way analysis of variance (ANOVA) for the results presented in the figures 1D and 3B-C (Table EV2). For the figure 3A, we performed a Tukey's HSD after multivariate analysis of variance (MANOVA; Table EV3). ...”

15. NCI Omics checklist, item 8: I believe that the intent of this item is to assess analytical performance of the assay and reported score rather than the clinical performance. It would be helpful to know the repeatability and reproducibility of the LC scores assessed on replicate samples from the same study participant, for example by display of Bland-Altman plots.

This is another excellent suggestion from Reviewer 1 that contributed to improve our work. In the revised version of the ms, we have determined the repeatability of isoform-specific expression analysis in EBCs using Bland-Altman plots. Due to space limitation of a Report at EMBO Mol Med, we briefly mention in the Results section of the main ms:

“...The repeatability of isoform-specific expression analysis of GATA6 and NKX2-1 was confirmed by Bland-Altman plots (Bland & Altman, 1986) after measurements in two EBCs from the same patient, in five patients (Fig EV4B, Appendix SupplementaryResults)...”

However, in the referred Fig EV4B and the Appendix Supplementary Results we elaborated more in detail this specific point as follows:

“...The repeatability of isoform-specific expression analysis of GATA6 and NKX2-1 was confirmed by Bland-Altman plots (Bland & Altman, 1986) after measurements in two EBCs from the same patient (Test 1 and Test 2), in five patients (Fig EV4B). For GATA6 (left panel), the mean of the differences between Test 1 and Test 2 of the five patients was 0.0155 with a 95% confidence interval (CI) from -0.004 to 0.0354 and limits of agreement between 0.028 and 0.056. Remarkably, all five differences in our experiment were within the limits of agreement. Furthermore four out of the five differences were within the 95% CI. Similar results were obtained for NKX2-1 (right panel), the mean of the differences between Test 1 and Test 2 of the five patients was 0.0035 with a 95% CI from 0.0006 to 0.0065 and limits of agreement between -0.003 and 0.001. All five differences were within the limits of agreement and three out of the five differences were within the 95% CI. These results confirmed the repeatability of isoform-specific expression analysis of GATA6 and NKX2-1 in EBCs...”

16. STARD checklist, item 13: The response given is "NA" but see comment #15.

We have changed the answer to the point 15 of the STARD checklist referring to the Fig EV4B and the corresponding text in the Appendix Supplementary Results.

17. STARD checklist, item 17: The response given is "NA" but it does seem relevant as some portion of patients had received treatment prior to collection of EBC sample.

We have changed the answer to the point 17 of the STARD checklist referring to the Table 2 in the main ms, in which is described the percentage of the population in each group receiving a treatment at the point of sample collection.

18. STARD checklist, item 18: The response given is "NA" but lung cancer stage, COPD, and IPF are all relevant.

We have changed the answer to the point 18 of the STARD checklist referring to the description of the Study population in the Materials and Methods section of the main ms.

19. If the completed STARD and REMARK checklists will be provided with the published paper, a better mechanism for indicating page number will be required as page numbering will change and it will be necessary to indicate whether the numbering refers to the main paper or one of the several supplements.

After acceptance of the manuscript, we will contact the editing team from EMBO Mol Med in order that the correct page numbering is given in the final version of the ms.

20. I did not attempt to run the R script on the data file provided in the Excel file, but it did not appear that the data were the same as indicated in the R script as the number of rows were different.

To facilitate the reproduction of our results using the R script, we have provided two additional files and included in the Materials and Methods section of the main ms the following description of how to use them:

“...Two files have been submitted for reproducing the results presented in the main figures and in the tables: an R Markdown file (GATA6_NKX2_1_EBC.Rmd) and a file containing the raw data (LC_Data.csv). (I) Both files have to be located into the folder, in which the output of the R Markdown should be saved. To run the R Markdown file, R version $\geq 2.15.0$ and R Studio are required. (II) After opening the R Markdown file with R Studio, the Knitr option should be selected. (IV) Using the raw data, the R Markdown will generate an html file containing all the plots from the main figures. In addition, several txt files containing the data from Tables 2, EV1-4, will be also generated...”

We would like to thank once again to Reviewer 1 for her/his constructive comments that led to a significant improvement of our ms.

Reviewer #3 (Remarks):

I like to congratulate the authors with their work and encourage them to proceed this research. It is an important upcoming research field where non-invasive diagnosis for lung cancer is being explored. Nevertheless, I have some comments and remarks about the manuscript.

We would like to express our gratitude to Reviewer 3 for his/her positive and the encouraging comments.

General comments

1. Please provide conflict of interest if there is any.

We would like to refer Reviewer 3 to the Disclaimer after the Authors contributions in the revised version of our manuscript (ms):

“The authors declared no conflict of interest. Pending patent applications PCT/EP2014/060489 (published as WO 2014/187881), EP 13 16 8629.7, EP 14 00 697.1, EP 14 19 5027.9”

2. In the significance section, it is stated that this method complements the success of CT: did you add the EBC results to the outcome of a CT scan in order to identify potential nodules as being malignant and where in the process would you place the EBC test: before or after the LDCT scan?

This specific sentence in the Significance section was written as a suggestion for the future. In any case, this section was removed from the ms. In the Discussion section of the revised ms, we have added new text describing more in detail how we propose to use the LC score to complement the success of CT:

“... We propose to incorporate our method into the current protocols for patients undergoing diagnostic evaluation for pulmonary diseases characterized by hyperproliferation. LC score might be also used to improve the sensitivity of risk prediction models for LC as previously suggested for biomarkers (Spitz et al, 2008). In addition, we suggest to integrate our technology into CT-based LC screening approaches in high risk populations (Bach et al, 2003; Cassidy et al, 2008; Colditz et al, 2000; de Torres et al, 2007; Spitz et al, 2007), a procedure routinely used in the US, but not in Europe due to concerns regarding the very high percentage of false positive observations (>90%) and hence low specificity (73.4%) (National Lung Screening Trial Research et al, 2013), resulting in unnecessary follow-up CT scans, bronchoscopy or even surgery (Jett, 2005). Concomitant implementation of EBC-based LC detection together with CT could help to reduce the false-positive rate of CT imaging, e.g. in cases with suspicious image findings, thereby preventing individuals from being unnecessarily exposed to high dose of radiation or surgery...”

3. Why do it in exhaled breath condensate and not in blood samples? It takes longer time to obtain exhaled breath condensate and are the methods reliable?

One of the main reasons to initiate this project using EBCs was the fact that we wanted to detect primarily lung diseases, thereby being EBCs closer correlated than blood samples. An additional reason is the future development of our method of LC detection using EBCs, which involves more advanced technology as the electronic Nose (eNose), which is a highly sensitive analysis platform for detection of diseases. Some of our co-authors are working in that direction.

Nevertheless, we would like to clarify that we are currently following this valuable suggestion from Reviewer 3. We initiated blood sample collection and measurements in parallel to EBCs analysis. However, these data will be the scope of future work.

4. Also, were all the patients able to give exhaled breath? Since the collection of exhaled breath condensate takes some time, patients need to sit with a nose-clip and breath through a mouthpiece. We see in the clinic that patients which are restricted have some trouble to perform this manoeuvre or experience this as oppressive.

Reviewer 3 is completely right with this specific comment. Not all of the patients (less than 2%) were able to breath for the minimal time of 5 min required for appropriate EBCs collection. All these patients were at advanced stage of LC. In these specific cases, the samples were not included in the study. However, for patients at initial stages of LC should not be a major problem to breath for 5 min through the cooling device for the proper EBC collection.

5. Did you take into account the recommendations for EBC sampling by the ERS/ATS task force (Horvath et al., European Respiratory Journal 2005; 26:523-548, DOI:10.1183/09031936.05.00029705)?

Yes, we did. In the revised version of the ms, we have included a sentence in the Material and Methods section stating:

“...EBC collection was performed using the RTube (Respiratory Research) as described online (<http://www.respiratoryresearch.com/products-rtube-how.htm>) and following the guidelines for EBC sampling by the ERS/ATS Task Force [16135737]...”

6. When reading the main article file for the first time, it felt overwhelming and rather confusing. A lot was explained in supplement. However, if possible, please restructure the text so that the main article file can be used as stand-alone article without the need for a supplement and in that way, supplement can contain extra figures or materials and methods, but important results should be included in the main manuscript. I felt 'lost' between main files, supplementary results and methods and figures.

We deeply appreciate this very constructive comment that helped us to improve significantly our ms. Following the suggestions from Reviewer 3, we changed the structure of the ms taking special care that the main text can be read and understood as stand-alone article. Important information for the main idea of the article was included into the main text, whereas information for specialists that required a deeper view into the data were included into the Expanded View Figures and Tables or in

the Appendix Supplementary Materials. We believe that the revised version of our ms will full fill the suggestions from Reviewer 3 allowing a better understanding of the paper and avoiding continues switching to the supplementary information.

Introduction (page 5)

7. It seems the reference from the Globocan website is missing. On the website is an example on how to refer to this website. Please provide a reference, with the date of consultation.

Thanks for the observation. The reference has been changed following the advices on the website.

8. Please provide a reference for the following sentence: "Approximately 55-85% of the CT-detected LC can be surgically removed resulting in high five-year patient survival of almost 52%."

A reference has been included

9. Is there any evidence for your postulation that many of the mechanisms involved in embryonic development are recapitulated during LC initiation. If so, please provide a reference.

Two references supporting this postulation have been included

10. What made you choose for GATA6 and NKX2-1 and not the other regulators in lung development as reported by Maeda et al., 2007?

Since a similar observation was also done by the other Reviewer (Reviewer 1), we decided to include the following text into the ms:

"...Thus, GATA6 (GATA Binding Factor 6) and NKX2-1 (NK2 homeobox 1, also known as TTF-1, Thyroid transcription factor-1), two transcription factors that are key regulators of embryonic lung development (Maeda et al, 2007), were analyzed for their potential as biomarkers for LC detection. Among other lung development relevant genes, *GATA6* and *NKX2-1* were selected due to their similar gene structure (Fig 1A) and their implication in LC (Cheung et al, 2013; Winslow et al, 2011)..."

11. In the last sentence, the results are already mentioned, which I feel should not be done here. Please rephrase your postulation so that it becomes a clear hypothesis.

We completely agree with Reviewer 3. The text has been changed following this advice.

Results

12. In the first phase, you optimized the SOP for RNA isolation out of FFPE samples. The second phase was the optimization of a SOP for EBC collection and storage, but then you used the SOP for RNA isolation out of FFPE material for the isolation out of EBC. Shouldn't an optimization be needed when collecting RNA from EBC or was the SOP exactly the same?

Please, allow us to explain the rationale behind these proceedings: The SOP for EBC-collection, -transportation, -storage and -processing (including RNA extraction) were optimized independent from the FFPE tissue samples due to the different characteristics of bot sample types. However, the SOP for assay operation (cDNA synthesis and qPCR amplification) were optimized using RNA extracted from FFPE tissue samples and later used on RNA extracted from EBCs, due to similar characteristics of the RNA isolated from both sample types: low amounts of highly fragmented RNA.

13. It seems table 1 is missing

We offer our apologies for this mistake. Table 1 has been included in the revised version of our ms.

14. In table S1, it seems that the number of current smokers have doubled in the control group of the validation set compared to in the training set (20.3% vs. 9.2%). Could this have influenced the results? Is a Chi-test used to look for significant differences between the groups?

In the revised version of the ms, all charts in the main figures are represented as box plots following suggestions from the other Reviewer (Reviewer 1). The five-number summary from each box plot and the statistical significance test values (including X^2) are presented in the Tables EV1-3. The results obtained after sample grouping based on the smoking history are presented in the Figure 3A and Table EV3 and described in the Results section of the main ms as follows:

“...Cigarette smoking is strongly associated with LC (Mehta et al, 2015b). Thus, we decided to assess whether the LC score reflected the smoking history of the individuals (Fig 3A). Controls and LC samples were sorted into three groups: never smokers (NS), previous smokers (PS) and current smokers (CS). The LC score was significantly different between Ctrl and LC patients in each of the smoking history groups (Table EV3). However, we did not find significant differences between the smoking history groups in the Ctrl nor in the LC group...”

In addition, we have incorporated into the main ms two tables (Fig 2E and Table EV4). These tables contains the positive predictive value (PPV), negative predictive value (NPV), true positive rate (TPR, sensitivity) and true negative rate (TNR, specificity) of the LC score when applied either to the validation set of EBCs (Fig 2E) or to sub-groups of our samples (Table EV4), for example excluding nonsmokers, or excluding patients with recurrent diseases, etc. In all the sub-groups the LC score achieved high performance values correlating with the values obtained after the external validation and arguing against potential bias related to sample distribution.

15. Controls? How were they chosen, were they followed prospectively and for how long? Was only EBC obtained? How could you be sure the healthy controls were healthy? How did you define COPD and IPF. Were there differences between smokers en non-smokers in the lung cancer group or between different stages of COPD, lung cancer? Were there differences between the lung cancer subtypes? Were there patients with COPD and lung cancer? Is there an influence of smoking (are there significant differences when comparing smokers to nonsmokers)?

For the questions related to smoking, we would like to feer the Reviewer 3 to the answer of the previous point.

For the rest of the questions in this specific point, we would like to attract the attention of Reviewer 3 to the description of the study population in the Materials and Methods section of the revised ms. In accordance with point 6 from Reviewer 3, we have tried to incorporate the most relevant information into the main text of the ms, in order that the ms can be read as stand-alone article. However, due to space limitations of a Research Report at EMBO Mol Med, we had to incorporate the detailed description of the study population into the Materials und Methods section. All the questions raised by the Reviewer 3 in this point are answered in the description of the Study population in the Materials and Methods section, including the definition of control samples, the time window for sample collection, the inclusion and exclusion criteria, the prevalence of the different LC subtypes in the study population, etc.

We respectfully hope that we were able to satisfactory answer this specific point from Reviewer 3.

16. Furthermore, it seems that the number of lung cancer patients overwhelms the numbers of controls, which could induce selection bias and hence, overoptimistic results. I suggest to include more controls in future work.

We offer apologies to Reviewer 3, but due to unpredictable complications beyond our control, we were not able to increase the number of age matching “healthy” donors in the time frame given for the revision of the ms. Nevertheless, to address the concern raised by Reviewer 3, we redefined the control population to include the samples from “healthy” donors together with IPF and COPD samples. The argumentation supporting the rational of this decision was mentioned in the Study population of the Materials and Methods section as follows:

“...The control population for the analysis of FFPE samples included lung tissue that was taken from macroscopically healthy adjacent regions of the lung of LC patients and control lung tissue that was obtained from age-matched donor lungs, who have had no diagnosis or family history of LC, in

the frame of surgical size reduction of the donor lung during lung transplantation. Lung tissue samples from IPF (idiopathic pulmonary fibrosis) and COPD (chronic obstructive pulmonary disease) patients, both diagnosed according to international guidelines (www.goldcopd.org) (Wuyts et al, 2012), were also included in to the control population because these lung diseases have been reported to increase LC risk when compared to individuals with normal pulmonary function (Li et al, 2014; Turner et al, 2007). In addition, the IPF and COPD cohorts were included in the study to determine the discriminatory power of the diagnosis method proposed here with respect to other non-cancer diseases characterized by alveolar or bronchiolar hyperproliferation...”

AND

“...The control population consisted of EBCs from healthy donors (22 individuals with no symptoms, no complaints and no prior history of LC or any other pulmonary disease), IPF and COPD patients (33 and 10 EBCs respectively). The rationale for including the IPF and COPD cohorts into control population was explained in the previous paragraph...”

The control population was representative of the LC population with respect to age, smoking history as well as gender distribution (Table 2). More importantly, performance assessment values of the LC score using a control population with (Fig. 2E) or without IPF and COPD samples (Table EV4) were similar arguing against potential bias by incorporating the IPF and COPD samples into the control population.

17. Why standardize to *TUBA1A*, while in supplement the genes *GAPDH* or *HPRT1* are used? Wasn't the same SOP used?

We assume that in this point there is a misunderstanding. All qRT-PCR expression results presented in the ms were normalized to *TUBA1A*. The expression analysis of *GAPDH* and *HRPT1* (in the revised ms Fig EV3E-F) using two different primer pairs for each of the transcripts was used to estimate the quality of the RNA. A similar method for RNA quality assessment was previously used (Fajardy I et al 2009). These results as well as the corresponding reference are described in the Appendix Supplementary Results as follows:

“...In order to ensure the reproducibility of the SOP proposed here, we employed as quality criterion for the mRNA purified from EBCs the ratio of expression of the housekeeping gene *GAPDH* (glyceraldehyde 3 phosphate dehydrogenase) detected using two different primer pairs that were complementary to different regions of the mRNA (Fig EV3E, top). To increase the stringency of quality assessment for RNA isolated from EBCs, we determined similar expression ratio of a second housekeeping gene, *HPRT1* (hypoxanthine phosphoribosyltransferase 1; Fig EV3F, top). Expression ratios of *GAPDH* and *HPRT1* close to 1.0 are indicators of high integrity of mRNA (Fajardy I et al 2009)...”

18. Are the Em/Ad ratio's reported as mean or median? Furthermore, the numbers mentioned (ratio's for controls for *GATA6* and *NKX2-1* of 0.256 and 0.186 and for lung cancer patients 1.792 and 2.228) seem not to be in accordance to figure 1B.

In the revised version of the ms, the Em/Ad expression ratios are reported as median. To avoid confusions we have improved the text in different places of the Results section by including the word “median”. As a representative example:

“...In control lung tissue ($n=61$), the median Em/Ad was 0.067 for *GATA6* and 0.143 for *NKX2-1* (Table EV1). Interestingly, the median Em/Ad increased in the LC tissue ($n=51$) to 2.247 ($P=3.446e-12$) for *GATA6* and to 1.617 ($P=2.04e-12$) for *NKX2-1*...”

19. Figure S2: how did you obtain healthy lung tissue? From the same patients (were the patients their own control), or from patients undergoing biopsies for other lung diseases (and are not "healthy")? This is mentioned in supplement but I think should be placed in the original manuscript file.

Please see the answer to the points 15 and 16.

20. Can you explain why Em/Ad ratio's dropped for *GATA6* in stage III lung cancer patients and why this is not the case for *NKX2-1*?

A potential explanation for the reduction of the Em/Ad ratio of *GATA6* at LC stage III might be that the expression of the Em-isoform of *GATA6* is important during LC initiation but not at later stages. In contrast, *NKX2-1* might be relevant during early and late stages of LC. However, these are assumptions that would need to be confirmed experimentally in the future.

21. Why isn't a plot shown after grouping by smoking status in Figure 1C?

We agree with Reviewer 3. This plot is important and it should be placed in the main figures. In the revised version of the ms you will find a box plot of the LC score in ctrl and LC patients after grouping based on the smoking history (Fig 3A).

22. How come there are no stage IV lung cancer patients in figure 1D when they are mentioned in table S1? Furthermore, why are only NSCLC patients shown and not SCLC?

We offer our apologies for these mistakes. This has been corrected in the new version of the plot in the Fig 1D.

23. Fig 2B: are there any explanations why there are some significant differences in patients between tissue and EBC ratio's?

Variation due to technical issues during preparation of the different types of samples from the same patient might be the most probable explanation for these discrepancies. However, we would like to insist that these discrepancies occurred in a minority of cases. In addition, the correlation between the results obtained from both sample types was relatively high for the Em/Ad ratios of both genes analyzed.

24. Figure S7C: the ROC curve shows very optimistic accuracy for the LC score. However, is the LC score the combination of both *GATA6* and *NKX2-1* scores or the SVM outcome? What exactly is meant with "point of operation"? Shouldn't this be the most upper left point?

After improving the structure of the ms following the suggestions from Reviewer 3 (see point 6.), the ROC curve analysis is now part of the main manuscript (Fig 2D). In this figure, the black solid line is the ROC curve for the LC score. The LC score is calculated by combining the Em/Ad expression ratios of *GATA6* and *NKX2-1* as indicated under "Classifier construction and LC score" of the Materials and Methods section. If we understood the question from Reviewer 3 correctly, our answer will be that the black solid line is the ROC curve for the SVM outcome.

Regarding the question to the "point of operation", we have changed the text in the corresponding Figure legend as follows:

"...The orange diamond represents the optimal operating point of the SVM classifier, the point on the curve with maximal Youden's J index...."

We respectfully hope that this text answer the question from Reviewer 3.

25. Figure S7: why are the ratio's log2-transformed and not log10-transformed like in figure S5?

In the Figure S5 were used also log2-transformed values. The mistake has been solved.

26. Page 11: the external validation was done on 68 controls. Are those only healthy controls or also include COPD/IPF controls? Please clarify in the text.

As described in the points 15. And 16., in the new version of the ms, the control population consist of "healthy donors", IPF and COPD samples.

27. Figure S8A-C: it is stated that the figure includes both the training and validation set. However, the training set should not be included since the validation set is independent. Same remark for figure S8-D. A training set is used to train a certain method and should not be used for validation. This will result in overly optimistic outcomes. If you want to have a larger number, you should include more patients in the validation set instead of combining validation and training set.

We completely agree with this point from Reviewer 3. During the restructuring of the manuscript, following the advice from Reviewer 3 (see point 6.), we decided to exclude the part of the manuscript in which we learned the SVM classifier using both sets of samples together.

28. How come that in figure 3, stage IV lung cancer is included and not in figure 1D?

Please see answer to the point 22.

29. In the figure caption of figure 3C, it is stated that data are represented as in figure 2B, to me this does not make any sense, since this is a totally different figure.

30. Figure 3D shows the separation of IPF, healthy controls and lung cancer patients. However, in figure S8 it is not shown to be able to discriminate IPF. How was the training done to discriminate IPF? Is the same thing done for COPD patients?

During the restructuring of the manuscript, following the advice from Reviewer 3 (see point 6.), we decided to exclude this part of the manuscript thereby increasing clarity and avoiding distraction from the main message of the ms.

31. Colour code in Figure S9 is not correct.

In the new version of the ms the charts of the previous Figure S9 are now presented as box plots in the main Figure 3.

32. In supplement material and methods it is mentioned that patients were sequentially included first in Mexico, then in Waldhof Elgershausen and finally in Giessen and Marburg. Because samples were not taken in parallel but sequentially, this could introduce bias due to differences location. Are the results robust enough and remain the same when a correction for location is done?

The legend has been changed according to the newly added box plots.

This might be a misunderstanding. We would like to confirm that samples were collected in parallel in all centers. However, all patients were recruited sequentially, means that they were incorporated into the study as they came to the clinic for diagnostic evaluation, representing a continuous patient recruitment. Thus, this does not mean that the samples were collected first in one cohort, and later in the other cohorts.

EBCs were prospectively collected in two chronologically separated sets, training set and validation set. However, EBCs of the same set (training or validation sets) were collected in parallel in the different cohorts.

For further details, we would like to refer to the “*Study design and study population*“ of the Materials and Methods section.

33. Further in supplement material and methods, the SOP for EBC collection is described. However, since this is an important factor in the study, I would suggest to include a part of this in the manuscript file. Furthermore, it is stated that every donor performed 10 minutes of tidal breathing but if this was too long, 5 minutes were also OK without loss in quality. Next to this, it is stated in the OMICS guidelines that EBC samples obtained in a time <5min were discarded and 10 minutes was optimal. However, when specifying the minimum amount of specimen needed, 5 min delivers approximately 500 μ l which is enough. So why wasn't 5 minutes used then as reference instead of the 10 minutes, which is still 5 minutes less for the patient?

We completely agree with this suggestion from Reviewer 3 regarding the importance of the description of EBC handling. We have included in the Material and Methods section of the main ms the following text:

“...EBC collection was performed using the RTube (Respiratory Research) as described online (<http://www.respiratoryresearch.com/products-rtube-how.htm>) and following the guidelines for EBC sampling by the ERS/ATS Task Force (Horvath et al, 2005). For EBC collection, it was advised that all donors refrain from eating and drinks (except water) for 2 hours before collection. Donors were awake and breathing normally without mechanical ventilation. Prior to EBC collection donors were asked to rinse the mouth with fresh water to avoid any additional contaminants. Sample was collected with the Rtube using a nose clamp to avoid nasal contaminants and breathing was only through the mouthpiece. For each donor, EBC collection was performed for 10 minutes of tidal breathing. However, if the donors felt any discomfort and/or inability to continue, a minimum time of 5 minutes was acceptable without any loss in quality of the material obtained. After EBC collection, the samples were stored immediately at -80°C in 500µl aliquots. It is essential that the samples are frozen as soon as possible after EBC collection (Fig EV3E-F). The EBC was stored in microcentrifuge tubes that were treated with Rnase Zap (Life technologies) and autoclaved twice. All steps during the collection and processing of EBCs were performed under RNase-free conditions, including the use of barrier-filter tips and cleaning all surfaces and gloves with Rnase Zap, which are critical to ensure the integrity and quality of the samples...”

To further answer the questions related to the time of EBC collection, a sample collection time of 10 min allowed us to have a surplus of material from the same patient that could be used as backup in case of unforeseen difficulties during the processing of the EBC. Furthermore, even if nothing went wrong, one could use the surplus for expression analysis of additional markers that could help to improve in the future the performance of the LC score.

34. In the OMICs guidelines (point 8) it is stated that an independent set of 79 previously unseen samples are used for external validation. However, in the text it states that 138 previous samples are used. Why this discordance?

It seems that we submitted the OMICs, STARD and REMAKR lists from a previous version of the ms. The mistakes have been solved by updating the lists and providing the version according to the revised ms.

Discussion

35. How do you see the future? The test as it is described now, it feels like a stand-alone diagnostic test. Can it be used that way? Furthermore, the last paragraph is attributed to the comparison with CT scans. But where would you put the test in the diagnostic work-up? And what should happen when the breath test is positive?

We have changed the Discussion section in order to address this specific point as follows:

“...We propose to incorporate our method into the current protocols for patients undergoing diagnostic evaluation for pulmonary diseases characterized by hyperproliferation. LC score might be also used to improve the sensitivity of risk prediction models for LC as previously suggested for biomarkers (Spitz et al, 2008). In addition, we suggest to integrate our technology into CT-based LC screening approaches in high risk populations (Bach et al, 2003; Cassidy et al, 2008; Colditz et al, 2000; de Torres et al, 2007; Spitz et al, 2007), a procedure routinely used in the US, but not in Europe due to concerns regarding the very high percentage of false positive observations (>90%) and hence low specificity (73.4%) (National Lung Screening Trial Research et al, 2013), resulting in unnecessary follow-up CT scans, bronchoscopy or even surgery (Jett, 2005). Concomitant implementation of EBC-based LC detection together with CT could help to reduce the false-positive rate of CT imaging, e.g. in cases with suspicious image findings, thereby preventing individuals from being unnecessarily exposed to high dose of radiation or surgery. Routine implementation of EBC-based molecular diagnosis may become an accurate, straightforward, non-invasive and low-price option to complement the success of CT for LC diagnosis...”

Materials and Methods

36. Was the study performed in accordance with the declaration of Helsinki? If so, please provide this in the study design section.

Yes. A corresponding sentence has been included into the description of the Study population of the Materials and Methods section.

37. In figure S1, in the start-1 phase you start with n=117. After excluding 5, you obtain n=151. How is this possible? Please provide more details on the number of samples.

We offer apologies for this mistake, which has been solved in the revised version of the ms. Tables 1 and 2 contains numbers related to the different sets of samples used in our ms. In addition, we would like to refer to the description of the study population in the Materials and Methods section.

38. It seems table 1 is missing

39. The reference to "R" is not correct.

40. What are the inclusion/exclusion criteria?

41. Please give more information about how the sample was taken, stored and shipped.

We offer our apologies for this mistake. Table 1 has been included into the ms.

The reference has been corrected according to the suggestion from the R core team.

Please, see the answers to the points 15. and 16.

We kindly attract the attention of Reviewer 3 to the description in Exhaled Breath Condensate Collection in the Materials and Methods section, which was also included as part of the answer of the point 33.

In addition, we also would like to refer to the description of the Fig EV3 in the Appendix Supplementary Results, subheading "*Optimization of qRT-PCR based expression analysis in exhaled breath condensate*", which contains results obtained during the optimization of EBC collection, storage, transportation and processing.

42. What p-value was considered to be the threshold for statistical significance?

P-values < 0.05 were considered as statistical significant. A corresponding sentence was introduced into the Statistical analysis of the Materials and Methods section.

43. The pages in the STARD checklist seem not to correspond and a lot of this important data is implemented in the supplement, which I think should be done in the main manuscript file.

It seems that we submitted the OMICs, STARD and REMAKR lists from a previous version of the ms. The mistakes have been solved by updating the lists and providing the version according to the revised ms.

44. Following STARD checklist, no intervals for sensitivity, specificity etc. are given.

Please see answer to point 43. In addition, to address this specific point, we have added to the ms Fig 2E, Table EV4 and Appendix Table S4.

45. In the REMARK checklist, a figure 4 is mentioned. However, through the manuscript, no figure 4 was shown or discussed. Please update the checklists.

We offer our apologies for this mistake, which have been solved as stated under point 43.

We appreciate the time and the effort expended during the review of our ms. We thank for the encouraging and motivating comments from Reviewer 3, as well as for her/his constructive

comments that led to significant improvements in our ms.

2nd Editorial Decision

12 September 2016

Thank you for the submission of your revised manuscript to EMBO Molecular Medicine. We have now received the enclosed reports from the referees that were asked to re-assess it. As you will see the reviewers are now globally supportive and I am pleased to inform you that we will be able to accept your manuscript pending the following final amendments:

- 1) Please take action on the final comments by the two reviewers.
- 2) We appreciate your suggestion for a synopsis image. However, could you provide a better quality image? I remind you that the final display size must be 550 px wide X (up-to) 400 px high.
- 3) Please provide separate "Results" and "Impact" sections in the "The Paper Explained" section.
- 4) I would suggest a more direct, impactful title with "diagnosis" in the fore. An example might be "Non-invasive, cost-effective lung cancer diagnosis by detection of GATA6 and NKX2-1 isoforms in exhaled breath condensate" or something similar. Please feel free to suggest alternatives.
- 5) Please do not modify the provided dataset formats and callouts in the manuscript (as you will see we created an EV6 dataset with appropriate callouts in the text; please check).

Please submit your revised manuscript within two weeks. I look forward to seeing a revised form of your manuscript as soon as possible. Do contact us if you have any doubts or questions.

***** Reviewer's comments *****

Referee #1 (Comments on Novelty/Model System):

The authors have provided a detailed account of the development of their novel EBC diagnostic test for lung cancer. They extensively revised the manuscript to address numerous comments from myself and another reviewer. The computations are made reproducible by the provision of R scripts and data. Relevant elements from three key checklists (STARD, REMARK, and OMICS) have been reported, greatly enhancing the completeness of the information provided.

My only remaining question is in regard to what I believe might be a typographical error. On page 13 there is reference to cohorts in America and Europe. I believe that the samples all came from Mexico and Europe (Germany only?), but the authors should clarify.

Referee #1 (Remarks):

My earlier comments have all been satisfactorily addressed. I recommend publication.

Referee #3 (Remarks):

I like to congratulate the authors for their excellent work and I really appreciated the reply on my comments. The manuscript has much improved and is much more readable since there is good structure to follow every step in the process.

However, there are some minor issues I like to suggest to adjust before the manuscript can be published.

First of all, I do not know if it is usual to give references in the abstract. If not, I would suggest to leave these out.

Furthermore, just for the clarity to the reader, please start the first sentence in the introduction by writing lung cancer in full.

Next, I would like to suggest the authors to be consistent in the way you represent the results. For instance in table 2, there are differences in the reporting of decimals and there are some blanks. As a general comment, I would suggest to do another grammar check.

2nd Revision - authors' response

23 September 2016

Please find as an attachment our revised manuscript (EMM-2016-06382-V2) with the title “Non-invasive lung cancer diagnosis by detection of GATA6 and NKX2-1 isoforms in exhaled breath condensate”. As you will confirm, we have done all amendments suggested in your mail from September 12, 2016:

1. You will find appended to this letter, a point-by-point response to the final comments of the Reviewers. Their suggestions were incorporated into the final version of our manuscript
2. The graphical abstract was improved following the guide lines for the authors from *EMBO Molecular Medicine*.
3. In the “The Paper Explained” section, the “Results” and “Impact” subsections were separated.
4. The title of the manuscript was changed following your kind suggestions.

As mentioned in my previous mail, we are very glad to read that the Reviewers as well as the Editorial Staff of your recognized journal confirmed the positive view about the novelty and quality of our work. We deeply appreciate the constructive editorial work from *EMBO Molecular Medicine*. If you require anything else for the publication of our manuscript, please do not hesitate to contact us.

Point-by-point response

Referee #1 (Remarks):

The authors have provided a detailed account of the development of their novel EBC diagnostic test for lung cancer. They extensively revised the manuscript to address numerous comments from myself and another reviewer. The computations are made reproducible by the provision of R scripts and data. Relevant elements from three key checklists (STARD, REMARK, and OMICS) have been reported, greatly enhancing the completeness of the information provided.

We would like to thank Reviewer 1 for the time and effort expended during the review of our manuscript. We also appreciate the positive comments from Reviewer 1.

My only remaining question is in regard to what I believe might be a typographical error. On page 13 there is reference to cohorts in America and Europe. I believe that the samples all came from Mexico and Europe (Germany only?), but the authors should clarify.

The text was changed according to the suggestions from the Reviewer 1.

My earlier comments have all been satisfactorily addressed. I recommend publication.

Thanks once again. Your suggestions guided us to improve significantly our work.

Referee #3 (Remarks):

I like to congratulate the authors for their excellent work and I really appreciated the reply on my comments. The manuscript has much improved and is much more readable since there is good

structure to follow every step in the process.

We appreciate the positive comments from Reviewer 3.

However, there are some minor issues I like to suggest to adjust before the manuscript can be published.

First of all, I do not know if it is usual to give references in the abstract. If not, I would suggest to leave these out.

In some journals is allowed to give references in the abstract. I would like to respectfully suggest that the Editorial Staff from EMBO Molecular Medicine decide whether these references should be removed or not from the abstract. I hope that you agree with this suggestion.

Furthermore, just for the clarity to the reader, please start the first sentence in the introduction by writing lung cancer in full.

The text was changed according to this suggestion from Reviewer 3.

Next, I would like to suggest the authors to be consistent in the way you represent the results. For instance in table 2, there are differences in the reporting of decimals and there are some blanks.

We followed this suggestion from Reviewer 3. In order to be consistent in the way of reporting the results we have tried to give 2 decimals in the complete manuscript. Only in few cases we moved away from this suggestion, reporting more or less than 2 decimals, with the aim to preserve the appropriate understanding of the manuscript.

As a general comment, I would suggest to do another grammer check.

The text was checked once again by other native speaker. Minor corrections were done. Thanks for this suggestion.

We would like to thank Reviewer 3 for the time and effort expended during the review of our manuscript.

Corresponding Author Name: Guillermo Barreto
Journal Submitted to: EMBO Molecular Medicine
Manuscript Number: EMM-201-06382